

# The Seasonal Phases of an Arctic Lagoon Reveal Non-linear pH Extremes

Cale A. Miller[1,3], Christina Bonsell[2], Nathan D. McTigue[2], Amanda L. Kelley[3]

[1] Department of Evolution and Ecology, University of California Davis, Davis, CA, USA, 95616
[2] Marine Science Institute, The University of Texas at Austin, Port Aransas, TX, USA, 78373
[3] College of Fisheries and Ocean Sciences, University of Alaska Fairbanks, Fairbanks, AK, USA, 99775

*Correspondence to:* Cale A. Miller (cmill@ucdavis.edu; calemiller620@gmail.com)



**Abstract**
The western Arctic Ocean, including its shelves and coastal habitats, has become a focus in
ocean acidification research over the past decade as the colder waters of the region and the
reduction of sea ice appear to promote the uptake of excess atmospheric $CO_2$. Due to seasonal
sea ice coverage, high-frequency monitoring of pH or other carbonate chemistry parameters is
typically limited to infrequent ship-based transects during ice-free summers. This approach has
failed to capture year-round nearshore carbonate chemistry dynamics which is modulated by
biological metabolism in response to abundant allochthonous organic matter to the narrow shelf
of the Beaufort Sea and adjacent regions. The coastline of the Beaufort Sea comprises a series of
lagoons that account for > 50 % of the land-sea interface. The lagoon ecosystems are novel
features that cycle between "open" and "closed" phases (i.e., ice-free, and ice covered,
respectively). In this study, we collected high-frequency pH, salinity, temperature, and PAR
measurements in association with the Beaufort Lagoon Ecosystem LTER for an entire calendar
year in Kaktovik Lagoon, Alaska, USA, capturing two open water phases and one closed phase.
Hourly pH variability during the open water phases are some of the fastest rates reported,
exceeding 0.4 units. Baseline pH varied substantially between open phase 2018 and open phase
2019 with a difference of ~ 0.2 units despite similar hourly rates of change. Salinity-pH
relationships were mixed during all three phases displaying no correlation in open 2018, a
negative correlation in closed 2018 – 2019, and positive correlation during open 2019. The high-
frequency of pH variability could partially be explained by photosynthesis-respiration cycles as
correlation coefficients between daily average pH and PAR were 0.46 and 0.64 for open 2018
and open 2019 phases, respectively. The estimated annual daily average $CO_2$ efflux was 5.9 ±
19.3 mmol m$^{-2}$ d$^{-1}$, which is converse to the negative influx of $CO_2$ estimated for the coastal



Beaufort Sea despite exhibiting extreme variability. Considering the geomorphic differences in
Beaufort Sea lagoons, further investigation is needed to assess if there are periods of the open
phase in which all lagoons are sources of carbon to the atmosphere, potentially offsetting the
predicted sink capacity of the greater Beaufort Sea.












## 1 Introduction

Acidification of the Arctic Ocean is predicted to proceed at a faster rate than lower latitude

regions due to the increased solubility of $CO_2$ in colder waters, intrinsically lower carbonate ion

concentration, and specific water mass mixing patterns (Fabry et al., 2009; Mathis et al., 2015).

The acidification phenomenon which increases the dissolved inorganic carbon to alkalinity ratio

reduces the natural buffering capacity of the carbonate system via a reduction in carbonate ion

concentration. These processes result in low calcium carbonate saturation state and a low sea

surface pH. It is estimated that the Canadian Basin, Beaufort Sea, and Chukchi Sea in the Arctic

have experienced a 2.7 % shoaling of low saturation state ($\Omega < 1.25$) waters from $0 - 250$ m over

the past 2 decades (Qi et al., 2017; Zhang et al., 2020). Future projections anticipate a

continuation of this trend with sustained, perennial, undersaturation of calcium carbonate in the

Beaufort and Chukchi Seas by the year 2040, which will reduce the capacity of these waters to

continually take up atmospheric $CO_2$ (Mathis et al., 2015). The rate at which this happens will

have significant implications on the current estimates of $CO_2$ uptake by the coastal Chukchi and

Beaufort Seas (Evans et al. 2015a). Acidification of offshore Arctic waters appear to be a

consequence of increasing Pacific Winter Water intrusion due to globally warming waters and an

influx of excess atmospheric $CO_2$ caused by the disequilibrium between air and seawater $PCO_2$

(Qi et al., 2017). Along the nearshore regions of the Beaufort Sea, however, coastal processes

predominately drive acidification such as riverine flux of freshwater, biological metabolism, sea-

ice melt from warming waters, and upwelling of the Polar Marine Layer which is an important

water source for Arctic lagoons (Carstensen and Duarte, 2019; Harris et al., 2017; Miller et al.,

2014; Woosley and Millero, 2020; Wynn et al., 2016).





The coastal margin of the Beaufort Sea consists of biologically complex, shallow (< 6 m),
discontinuous, estuarine lagoons that depict ∼ 50 % of the coast from Nuvuk (Pt. Barrow) to
Demarcation Bay, Alaska, USA (Dunton et al., 2006, 2012; Harris et al., 2017; Lissauer et al.,
1984). The North Slope region is predominately tundra, where the annual terrestrial thaw
comprises the majority of the freshwater outflow to the Beaufort Sea. Canada's Mackenzie River
is the largest source of freshwater flowing into the Beaufort Sea, ∼ 300 km$^3$ yr$^{-1}$ (McClelland et
al., 2006; Stein and Macdonald, 2004); however, many smaller rivers and streams link the
terrestrial hydrography with the marine lagoon ecosystem characterized as geomorphic transition
zones (Dunton et al., 2006, 2012). Barrier islands partially obstruct Beaufort Sea coastal water
exchange with the lagoons, which in part are hydrographically influenced by the seasonal shifts
in terrestrial freshwater flux that results in highly dynamic chemical conditions (Mouillot et al.,
2007). Flow channels between the land, Arctic lagoons and the ocean are ephemeral, causing the
flow of water in and out of a lagoon to be intermittent, varying on short- and long-term time
scales (Dunton et al., 2012; Kraus et al., 2008). These physical flow attributes result in highly
variable salinity and temperature that range from fresh to hypersaline (0 to >45), and -2 ˚C to 14
˚C, respectively (Dunton and Schonberg, 2006; Harris et al., 2017). This variability in
temperature and freshwater delivery can have a dramatic effect on carbonate chemistry
thermodynamics and modify alkalinity and dissolved inorganic carbon (DIC). The seasonality of
these shallow lagoons is distinguished by two principal phase states corresponding to sea ice
prevalence—open and closed. The closed period during winter ice cover exhibits a non-
quantifiable amount of air-sea exchange due to the physical sea ice barrier. Conversely, the open,
ice-free summer period from late spring to early fall is marked by spring river discharge, air-sea
exchanges, and meteorological events (McClelland et al., 2012, 2014). Episodic fluctuations in



lagoon hydrography during periods of open water add to the complexity of physicochemical
variability as wind-driven upwelling events coupled with tidal flux can precipitate rapid changes
in these semi-isolated bodies of water (Lissauer et al., 1984).

Despite extreme variability in temperature and salinity, Arctic lagoons are home to

diverse fish assemblages that include diadromous, freshwater, and marine species (Harris et al.,
2017; Robards, 2014; Tibbles, 2018), many of which serve as important subsistence fisheries for
Arctic communities (Craig, 1989; Griffiths et al., 1977). Arctic lagoons have relatively high
diversity and abundance of benthic community invertebrates, ranging from 654 to 5,353
individuals m$^{-2}$ with trophic linkages to birds and marine mammals (Griffiths et al., 1977,
Johnson et al., 2010; Dunton et al., 2012). The benthic food web relies on both autochthonous
microalgal production and allochthonous terrestrial organic matter (OM) inputs as carbon
subsidies (Harris et al., 2018). The deposition of these carbon subsidies may have implications
on the chemical conditions of lagoon ecosystems via enhanced remineralization during the
during open and closed phases. To date, hydrographic physicochemical measurements have been
mostly limited to the open [summer] season with few exceptions (Dunton and Schonberg, 2006;
Kinney et al., 1971; Mathews and Stringer, 1984; Robards, 2014). To our knowledge, only a
single high-frequency year-round measurement of Beaufort Sea lagoon temperature and salinity
exists (Harris et al., 2017), which is insufficient for understanding how these factors including
biological metabolism may impact carbonate system dynamics.

This study is the first to incorporate a high-frequency time series of salinity, temperature,

PAR, and pH for an entire calendar year capturing both open and closed phases of an Arctic
lagoon. The Kaktovik Lagoon located adjacent to Barter Island and the city of Kaktovik was
selected for sensor package deployment. The data collected in this study were processed in part





with those available from the Beaufort Lagoon Ecosystems (BLE) Long Term Ecological
Research Program (LTER) and the NOAA Earth Systems Research Laboratory (ESRL). Salinity,
temperature, and pH were analyzed in the time and frequency domains alongside ancillary solar
radiation and water depth in order to examine potential modifiers of pH. This included estimates
of carbon flux at the land-sea interface utilizing atmospheric $PCO_2$ measurements and comparing
those with derived seawater $PCO_2$ estimates. The findings of this study are presented in the
context of seasonal variability of oceanographic processes in an ecosystem that is part of the
western coastal Arctic that is experiencing climate change.

**2   Study site and methods**
**2.1   Kaktovik Lagoon ecosystem**
Kaktovik Lagoon, Alaska (70° 6' 3" N 143° 34' 52" W), serves as one of the study sites for the
National Science Foundation's Beaufort Lagoon Ecosystem (LTER). It is one of a series of
coastal lagoons that fringe the Arctic National Wildlife Refuge and borders the east side of
Barter Island. With a maximum depth of approximately 4.4 m, Kaktovik Lagoon has two narrow
exchange pathways with adjacent water bodies (Dunton et al., 2012). One of the pathways
connects to Arey Lagoon, the other links to Jago Lagoon and to the Beaufort Sea via a channel >
25 m long and < 2.5 m deep (Fig. 1). Surface freshwater inputs are limited to small tundra
streams, although narrow inlets provide some exchange to adjacent Arey and Jago Lagoons,
which receive terrestrial inputs from the Hulahula/Okpilak and Jago Rivers, respectively. The
timing of sea ice formation varies by year but occurs between late September and October
becoming landfast (fastened to the coastline) in the shallow lagoons until breakup in May or June
(Dunton et al., 2006).






## 2.2 Oceanographic sampling

A benthic mooring outfitted with a SeaBird SeaFET V2 and RBR Concerto CTD++ was

deployed 8 August 2018 to 11 August 2019, with sensors roughly 10 cm from the bottom in

Kaktovik Lagoon (Fig. 1). Hourly measurements of pH, salinity, and temperature (from SeaFET

thermistor) were recorded (UTC) throughout the deployment period. A separate, adjacent

mooring consisting of a LI-COR spherical quantum sensor in-line with a LI-1000 datalogger

recorded photosynthetically active radiation (PAR μmol photons $m^{-2}$ $s^{-1}$; 400-700 nm) ~30 cm

from the bottom. Average PAR was integrated over three-hour time periods and recorded. In

April, August, and June, the site was sampled for dissolved nutrients and physicochemical

parameters within 30 cm of water surface and within 30 cm of the bottom. Physicochemical

parameters were recorded with a YSI ProDSS calibrated daily before excursions. Nutrient

samples were collected with a peristaltic pump fitted with Masterflex C-flex tubing, then filtered

through a Geotech 0.45 μm high-capacity polyethersulfone (PES) capsule filter connected with

Masterflex-C tubing and frozen at -20 ˚C until analysis. Sediment was retrieved from the

seafloor by a 0.1 $m^2$ van Veen grab, sampled with 50 mL push core and frozen at -20 ˚C until

analysis. Porewater was extracted by centrifugation of defrosted sediment, then analyzed

immediately. Dissolved nutrients in water and porewater [ammonia ($NH_3$), nitrate + nitrite

($NO_x$), orthophosphate ($PO_4^{3-}$), and silica ($SiO_2$)] were measured at the Core Facilities

Laboratory at The University of Texas Marine Science Institute in Port Aransas, Texas on a

continuous flow-analyzer Lachat Quick Chem 8500.


## 2.3 Seawater chemistry and sensor calibration



Discrete bottle samples were taken approximately 10 cm off the bottom proximal to the sensor
on 17 August 2018 for SeaFET calibration, and 26 April 2019 for reference. Bottle samples were
collected in duplicate and processed for total alkalinity-$A_T$ and $pH_T$ (total scale). An additional
$A_T$ sample was collected on 21 June 2019. The August 2018 sample was gathered by Van Dorn
bottle, where a single sampling was used to fill duplicate bottle replicates. April 2019 duplicate
samples were directly collected from depth by a peristaltic pump fitted with MasterFlex C-flex
tubing. All seawater samples were placed in 500 mL borosilicate bottles and fixed with 200 μL
saturated mercuric chloride and held at 4 ˚C until laboratory analysis.

$A_T$ was measured with an open-cell titrator using 0.1 M hydrochloric acid titrant on a

Metrohm Titrino 848 (Dickson et al., 2007: SOP 3b). Spectrophotometric $pH_T$ measurements
were made in duplicate using a Shimadzu 1800 outfitted with a cuvette temperature controller
stabilizing temperature at 25 ˚C. The spectrophotometric $pH_T$ was determined using *m*-cresol
purple (Acros, batch # 30AXM-QN), following SOP 6b from Dickson et al. (2007). An impurity
correction factor of the *m*-cresol reagent was used to adjust the final measured $pH_T$ value
(Douglas and Byrne, 2017). All salinity measurements were conducted with a YSI 3100
conductivity meter. Certified Reference Material of seawater (CRM: Batch 172, A.G., Dickson,
Scripps Institute of Oceanography) was used to calculate the $A_T$ and *m*-cresol dye uncertainty.
Calibration and reference *in situ* $pH_T$ samples were derived using the Matlab version of CO2SYS
(van Heuven et al., 2011) with input parameters salinity, temperature, $pH_T$, and $A_T$ using
dissociation constants from Lueker et al. (2000), Dickson et al. (1990), and Uppström (1974).

A SeaFET conditioning period of 9 d was conceded from deployment on 8 August 2018

to 17 August 2018 when the calibration sample was collected. A single-point calibration was
applied following previously established best practices (Bresnahan et al., 2014; Miller et al.,



2018). New calibration coefficients for the SeaFET were then applied and used to calculate $pH_T$
from the internal ISFET electrode for the entire dataset (Martz et al., 2010). The single reference
sample taken on 26 April 2019 was used to compare against SeaFET measured $pH_T$ as a check
for sensor drift and robustness of calibration.

**2.3.1 Uncertainty estimate**
The reliability and accuracy of SeaFET sensors is dependent on estimating the total uncertainty
attributable to an individual sensor's behavior and operator usage (Bresnahan et al., 2014;
Gonski et al., 2018; McLaughlin et al., 2017; Miller et al., 2018; Rivest et al., 2016). A previous
method for calculating the total uncertainty associated with SeaFET function has been previously
proposed and was applied to this study (Miller and Kelley 2020 *in review*). Briefly, a propagated
uncertainty Eq. (1) was derived by adding in quadrate the standard deviation of analytical
replicates measuring CRM $pH_T$ spectrophotometrically, a titrator uncertainty comparing
measured and known $A_T$ from CRM, the standard deviation of discrete $pH_T$ bottle replicates, and
the uncertainty associated with CO2SYS dissociation constants using the Matlab errors function
described in Orr et al. (2018). An additional salinity uncertainty not described in Miller and
Kelley (2020 *in review*) was added to account for the discrepancy between benchtop salinity
measurements and *in situ* readings found in this study (Table S1). The final equation reads:
$$Q = \sqrt{\sigma^2_{m-cresol} + \sigma^2_{bottle\ replicates} + \sigma^2_{CO2SYS\ constants} + \sigma^2_{salinity} + AN^2_{titrator}} \quad (1)$$
where $Q$ is the propagated uncertainty, $AN$ is the anomaly between measured and known $A_T$, and
$\sigma^2$ is the standard deviation of all of the uncertainty input parameters in pH units (see Miller and
Kelley 2020 *in review*). From this point, the total uncertainty was calculated by taking the



average of the propagated uncertainties for the calibration sample, reference sample, and bottle
anomaly (Table 1). This propagated uncertainty was then applied to the entire $pH_T$ time series.

**2.4   Ancillary data acquisition**
The Beaufort Lagoon Ecosystems LTER data on current velocity, water depth, and underwater
PAR was accessed through the Environmental Data Initiative portal. Current velocity was used
as a proxy to determine the open and closed (i.e., ice covered or ice-free) seasons for the lagoon.
A velocity consistently below 2 cm s$^{-1}$ for a period >10 h was designated as a threshold for the
two phases (Fig. S1). Water depth derived from the pressure sensor was interpreted as tidal
variation, where consistent frequencies in depth changes were applied for analysis (see 2.5).
Instantaneous PAR measurements were used to determine daily average values for time series
analysis.

**2.5   Frequency Analysis**
A power spectral density (PSD) analysis of $pH_T$, temperature, salinity, and tide was performed
using the *pwelch* function in Matlab (v2020a) to determine the magnitude of variation at a given
frequency during each phase: open 2018, closed 2018 – 2019, and open 2019. This function
processes data as samples s$^{-1}$, so for 24 measurements in a day, a sampling rate of 2.78 x 10$^{-4}$ was
applied with a frequency of d$^{-1}$. A Hamming window was used for sidelobe attenuation of the
analyses and the mean value for each parameter was subtracted in order to examine only the
variation around the mean. Residual noise around a frequency of 0 was muted by applying a
Butterworth high-pass filter with an order of 3 and cut off frequency at 1.0 x 10$^{-5}$. If two of the
analyzed variables exhibit the same predominate frequency, then their variation is assumed to be



correlated regardless of direction and magnitude. Previous PSD analyses with similar parameters
have been shown to be considerably noisy below ~ 50 dB Hz$^{-1}$, thus making this value a cutoff
threshold for the purposes of this study (Miller and Kelley *in review*).

**2.6   $A_T$, $PCO_2$, and flux calculations**
Salinity recorded by the RBR Concerto CTD++ were filtered for invalid measurements taken
over the year-long time series. Measurements identified as below the freezing point of water due
to the temperature-salinity relationship were removed, and a linear interpolation was performed
to replace the missing values (Fig. S2). Two linear regression analyses were performed to
estimate $A_T$, one with measured *in situ* salinity and the other with benchtop recorded values.
Each analysis was constructed with the three discrete $A_T$ samples collected on 17 August 2018,
26 April 2019, and 21 June 2019 (Table S1), where $A_T$ is the dependent variable and salinity the
independent. Benchtop values were considered to be more robust as the YSI 3100 Conductivity
meter was calibrated to the manufacturer's specification, while the CTD++ was factory
calibrated. For this reason, the regression from the benchtop salinity measurements were
considered to be the primary hourly $A_T$ values; however, both $A_T$ estimates from benchtop (slope
= 59.71, R2 = 0.968) and *in situ* (slope = 48.38, R2 = 0.998) salinity were used as input
parameters along with measured $pH_T$ to calculate hourly $PCO_2$ values using CO2SYS (see above
for constants applied).

Atmospheric hourly $PCO_2$ averages were collected from the NOAA ESRL station at

Barrow (Utqiaġvik), Alaska, USA (Thoning et al., 2020), and wind speed was acquired from
automated airport weather observations from the Barter Island Airport. Using these data,





a $CO_2$ air-sea flux for open phases 2018 and 2019 was calculated following the bulk transfer
method with a gas transfer velocity constant $k$ as modified by the Schmidt number (i.e., ratio of
kinematic viscosity of water to gas diffusivity), which is a function of temperature and salinity.
The bulk flux equation in Wanninkhof (2014) was used for the estimate:

$$F_{bulk} = 0.251 < U^2 > (Sc/660)^{-0.5} K_0 \, (PCO_{2_w} - PCO_{2_a}) \qquad (2)$$

where $U$ is wind speed in m s$^{-1}$, $Sc/660$ is the Schimdt number calculated using the coefficients
from the 4$^{th}$ order polynomial in Wanninkhof (2014: Table 1), $K_0$ is temperature and salinity
dependent solubility of $CO_2$ in mol L$^{-1}$ atm$^{-1}$ calculated following the model presented in
Wanninkhof (2014: Table 2), and $PCO_2$ is the partial pressure of $CO_2$ in water ($_w$) and air ($_a$) in
atm. Since the Schimdt number is a function of temperature and salinity, a freshwater value was
derived using the *fw* coefficients presented in (Wanninkhof, 2014). This estimate provided a
more conservative flux and was, therefore, presented as the lower bound uncertainty in the
estimate. The upper bound uncertainty of the flux estimate was calculated by applying the $PCO_2$
values into Eq. (2) derived from the salinity$_{in\ situ}$-A$_T$ regression. These values resulted in a larger
flux estimate, which is why they were set as the upper bound. Both the lower and upper bounds
were then applied as the total uncertainty for the flux estimate.

**2.6  Statistical applications**
Relationships between pH$_T$ and salinity were correlated by applying a quadratic fit for the closed
2018 – 2019 phase and open 2019 phase with salinity as the explanatory variable. No correlation
existed for open 2018. pH$_T$ and PAR hourly variations were collapsed by calculating the daily



averages for both parameters. The average daily values for pH$_T$ open 2018 and 2019 were then
detrended to remove correlations with salinity. A Pearson's correlation coefficient was than
derived between the detrended pH$_T$ daily averages and PAR daily averages for open 2018 and
open 2019.

**3   Results**
**3.1   Time series**
The year-long time series of pH$_T$, temperature, and salinity was recorded from 17 August 2018 to
11 Aug 2019 (Fig. 2). Based on the current velocity threshold of 2 cm s$^{-1}$ as a proxy for sea ice
cover, the 2018 open phase transitioned to a closed phase on 8 October 2018 which terminated
on 22 June 2019 as the 2019 open phase began (Fig. S1). Both calibration and reference samples
that were collected in duplicate have a fairly high standard deviation at 0.099 and 0.088,
respectively. The large deviation between duplicate samples was the greatest source of
uncertainty (see Eq. 1) for the entire pH$_T$ time series, which shows the total uncertainty shaded in
grey (Fig. 2a) and found in (Table 1). Invalid salinity values were ~ 6 % of the entire time series,
with the greatest proportion of interpolated values concentrated in the closed phase (Fig.2c).

In the open phase of 2018 pH$_T$ values were highly variable in August ranging from 7.66

to 8.40, which was the highest pH$_T$ recorded for the entire calendar year (Fig. 3a). An upward
trend in pH$_T$ began on 21 August and steadily increased indicating a continued accuracy of the
internal ISFET at low salinity. The low episodic salinity event when values were < 9 occurred
from 23 August to 27 August 2018, which was after the sporadic variability in pH$_T$ days earlier
(Fig. 3). From September until freeze-up on 8 October, pH$_T$ variability was low with the 7-d
running average maintaining at ~ 8.10 and fluctuating < 0.1 units. Temperature followed a steady



decrease with a negative slope of 0.12 (Fig. 3b). Salinity rose steadily although instances of large
episodic events were present, and in one instance on 1 September, salinity increased from 12.9 to
23.1 in an 8 h period (Fig. 3c).

During the closed phase when Kaktovik Lagoon first became ice-covered, $pH_T$ continued

to remain somewhat invariant around ~ 8.10 as it did during the previous two open-water months
(Fig. 4a).  Approximately 2 weeks into the closed phase, $pH_T$ began to steadily decrease until
stabilizing in the beginning of January at ~ 7.71.  $pH_T$ varied between 7.55 and 7.85 from this
point until April when another negative trend culminated at a low of 7.48. Late May saw $pH_T$
levels increase until phase transition on 22 June 2019. Temperature stayed below -1 ˚C until late
May when it began to increase concomitantly with $pH_T$ approaching 0 ˚C (Fig. 4b). Salinity
values increased from 31 at the start of ice cover reaching a maximum of 39.2 in April (Fig. 4c).

Open phase 2019 saw extreme $pH_T$ variability beginning 21 June to 11 August 2019 with

the rate of hourly change reaching as high as 0.467 units (Fig. 5a). During the first portion of this
phase, the $pH_T$ running average was consistent at ~ 8.05 and shifting only ± 0.05 units. Episodic
fluctuations caused $pH_T$ values to reach as high as 8.33. A negative trend began in late July
shifting the running average to ~ 7.79, which was ~ 0.2 units lower than the running average in
August 2018. Temperature increased rapidly during the first 2 weeks following break up and
then remained stable around 10 ˚C (Fig. 5b). Salinity decreased steadily for the first month after
break-up followed by large episodic freshening events in late July (Fig. 5c); these were similar to
the events seen in the open phase of 2018.

Correlations between salinity and $pH_T$ were inconsistent and varied by phase. Open phase

2018 $pH_T$ was not correlated with salinity which ranged from 5 to 30, while $pH_T$ was
predominantly steady shifting only ± 0.1 units around 8.0 (Fig. 6a). The maximum range of $pH_T$



during this period was confined to salinity values between 11.5 to 12.5. During the closed phase,
$pH_T$ correlated well with salinity, which ranged from ~ 30 to 40 (Fig. 6b). An inverse
relationship between salinity and $pH_T$ was present during this phase with an $R^2$ of 0.69. The
opposite pattern was observed during open phase in 2019, however, where salinity and $pH_T$ were
positively correlated with an $R^2$ of 0.66 (Fig. 6c). The temperature relationships with salinity
were due to seasonal timing rather than intrusion of water mass or mixing.

**382    3.2   Frequency of pH variability**

The PSD of $pH_T$ during open phase 2018 and closed phase 2018 – 2019 were weak with the
majority of peaks around any given frequency falling under 50 dB $Hz^{-1}$ (Fig. 7a and b). Peaks of
$pH_T$ during open 2018 did not correspond with any regular frequencies across temperature,
salinity (Fig. 7) or tide (Fig. S3), which only displayed regular peaks at a frequency of 1 and 2 $d^-$
$^1$. Consistent variability of $pH_T$ during the closed phase was negligible but had a maximum
magnitude at a frequency of 0.39 which corresponded to a peak observed with temperature (Fig.
7b and e). Open phase 2019 had a multitude of peaks with frequencies ranging from 0.5 to 7.5 $d^-$
$^1$, however most fell under 50 dB $Hz^{-1}$ (Fig. 7c). The highest magnitude of $pH_T$ corresponded
well with tide at ~ 1 $d^{-1}$ (Fig. 7c and S3c). Salinity also displayed a strong peak at 1 $d^{-1}$ (Fig. 7i),
sharing this frequency of variability with $pH_T$ and tide.

**394    3.3   pH response to PAR**

Open phase 2018 and open phase 2019 daily average $pH_T$ was compared against instantaneous
underwater PAR levels recorded for both phases (Fig. 8). Open phase 2018 PAR levels were
consistently lower compared to open phase 2019 as a result of the time of year the two phases



were observed (Fig. 8). The detrended daily average $pH_T$ correlated well with daily average PAR
with a Pearson's correlation coefficient of 0.469 (*p-value* = 0.005). In early August 2018, PAR
levels > 5 μmol photons $m^{-2}$ $s^{-1}$ were not representative of, high, daily average $pH_T$. This was a
deviation from the general trend of the open 2018 phase in which daily average $pH_T$ was
positively correlated with instantaneous PAR (Fig. 8a). In late August and September, high
values of daily average $pH_T$ > 8.20 coincided with spikes in instantaneous PAR that exceeded 10
μmol photons $m^{-2}$ $s^{-1}$ (Fig. 8a).

Open phase 2019 daily average $pH_T$ was overall more variable than open phase 2018

with values from 7.66 in early August to 8.09 in late June (Fig.8b). The detrended daily average
$pH_T$ had a more robust correlation with daily average underwater PAR than in 2018 with a
Pearson's correlation of 0.643 (*p-value* < 0.000). The highest PAR values were recorded in mid-
July; however, this did not correlate with the highest daily average $pH_T$ which was observed in
late June. Consistent high values of PAR in mid-July corresponded to relatively flat daily
average $pH_T$ (Fig. 8b). A reduction in instantaneous PAR to values below 15 μmol photons $m^{-2}$ $s^{-}$
$^{1}$ in late July was linked with a gradual decrease in daily average $pH_T$. During this 11-d period,
daily average $pH_T$ dropped from 8.06 to 7.71, and only began to increase again when
instantaneous PAR exceeded 25 μmol photons $m^{-2}$ $s^{-1}$ for consecutive days.

**3.4   Flux Estimation**
Carbon flux estimates for open phase 2018 and open phase 2019 showed dramatically different
results with 13 instances exceeding a flux > 10 μmol $CO_2$ $m^{-2}$ $min^{-1}$ compared to 302 instances in
open phase 2019 (Fig. 9)—where 10 μmol $CO_2$ $m^{-2}$ $min^{-1}$ is ≈ to 2 mmol $CO_2$ $m^{-2}$ $d^{-1}$ which is the
equivalent magnitude, but opposite of the estimated annual mean sea-air flux for the coastal



Beaufort Sea, -2 mmol $CO_2$ m$^{-2}$ d$^{-1}$ (Evans et al, 2015a). The episodic events of flux from the
atmosphere into seawater was greater in 2018 with 21 instances < -10 µmol $CO_2$ m$^{-2}$ min$^{-1}$
compared to a single instance in 2019. The maximum lower bound uncertainty for open phase
2018 was estimated at 2.23 µmol $CO_2$ m$^{-2}$ min$^{-1}$ whereas the upper bound was 10.67 µmol $CO_2$
m$^{-2}$ min$^{-1}$ (Fig. 9a). Overall, wind speed correlated poorly with $CO_2$ flux in 2018 ($R^2 = 0.13$). The
highest frequency of robust wind speeds occurred in October but resulted in only a minor
atmospheric flux into seawater as the majority of values were between 2 and -5 µmol $CO_2$ m$^{-2}$
min$^{-1}$ (Fig. 9a).
Open phase 2019 had an estimated $CO_2$ flux as high as 105 µmol $CO_2$ m$^{-2}$ min$^{-1}$, which
occurred in early August (Fig. 9b). Over a 5.6 d period in late July, $CO_2$ flux was > 10 µmol $CO_2$
m$^{-2}$ min$^{-1}$ for more than 90 % of the time reaching a high of 78 µmol $CO_2$ m$^{-2}$ min$^{-1}$. The
maximum lower bound uncertainty estimate for open phase 2019 was 5.5 µmol $CO_2$ m$^{-2}$ min$^{-1}$
with an upper bound uncertainty of 8.56 µmol $CO_2$ m$^{-2}$ min$^{-1}$. Wind speed was found to be
significantly correlated with $CO_2$ flux (*p-value* < 0.0001, $R^2 = 0.53$) in 2019 and, thus, cogently
different from open phase 2018.

**4   Discussion**
Kaktovik Lagoon was an ideal location for a year-long deployment to capture the three phases
(i.e., open 2018, closed 2018 – 2019, and open 2019) of environmental conditions in the coastal
Arctic. The study site displayed annual pH variability in the context of a unique lagoon where
geographical and physical features of this site represent a semi-closed system with narrow
passages to the sea and only small tundra stream inputs. The stochastic events of pH captured in
this system are some of the most dramatic hourly pH rates of change recorded to date (Cyronak



et al., 2020; Hofmann et al.; 2011; Kapsenberg et al., 2015; Kapsenberg and Hofmann, 2016;
Takeshita et al., 2015). These findings represent a system that is often in tenuous equilibrium
resulting in dramatic fluctuations of $CO_2$ outgassing and differing magnitudes of pH sensitivity
to temperature and salinity. The extreme nature of these habitats displays the resilience of the
micro and macro faunal community that undoubtedly modify seawater pH via biological
processes. While this study was able to capture physical and chemical conditions of the lagoon,
future work should be directed toward understanding how community organization in the lagoon
ecosystem affect pH variability.

**4.1   Kaktovik Lagoon and pH-salinity relationship**
A crucial finding from this year-long time series was the disparity between the $pH_T$-salinity
relationship during the open 2018, closed 2018 – 2019, and open 2019 phases. Sequentially
through the time series, the $pH_T$-salinity relationship was non-existent, negatively correlated, and
positively correlated, indicating that multiple processes drive pH variability at differing
magnitudes at a seasonal-phase resolution. Given the myriad processes such as temperature-
salinity relationships with carbonate chemistry, current- and wind-driven flux between the
sediment-water interface and the air-sea interface, as well as photosynthesis and respiration
cycles (Carstensen and Duarte, 2019; Hagens et al., 2014; Rassmann et al., 2020; Zeebe and
Wolf-Gladrow, 2001), it is unsurprising that salinity was observed as only a moderate and
intermittent driver of $pH_T$ variability in Kaktovik Lagoon. This is despite the multitude of
salinity changes that shift in time due to the discharge from rivers and tundra streams, seasonal
ice-formation and break up, and water column stratification, all which would be expected to
fluctuate pH predictably. The features intrinsic to Kaktovik Lagoon are likely important factors





responsible for the degree of $pH_T$-salinity interdependence and provide a lens that elucidates $pH_T$
altering processes that are less germane to physical oceanographic open-ocean mechanisms such
as temperature and salinity.

The characteristics of the Beaufort Sea lagoon ecosystems are unique features of the

coastline and exist as an interface between terrestrial inputs and seawater with each lagoon
varying in its connectivity to the Beaufort and freshwater sources. These lagoons temporarily
trap large amounts of allochthonous particulate organic carbon—which is expected to increase
with warming temperatures—and sediment as river and stream discharge are temporarily
mismatched between spring freshet and ice-covered margins (Dunton et al., 2006; Schreiner et
al., 2013). The lagoons adjacent to Kaktovik (Arey and Jago) are likely to be more exogenously
influenced due to greater connectivity to the Beaufort Sea, and the Okpilak, Hulahula, and Jago
Rivers. Thus, the modification of $pH_T$ within Kaktovik Lagoon provides a baseline that is likely
dissimilar to adjacent lagoons providing an in-depth examination of the internal processes of a
"closed system" such as biological metabolism and sediment flux that can drive seasonal pH
variability and explain the annual shifts in moderate salinity dependence.

In the open phase of 2018, $pH_T$ values were observed to be > 8.05 despite the striking

range of salinity from 5 to 30. This included an event that modulated salinity from 13 to 23 over
an 8 h period, which was correlated with high NW winds at ~ 20 m s$^{-1}$. This suggests that higher
salinity waters from the adjacent Arey Lagoon connecting the Beaufort Sea may have mixed into
the bottom waters were the pH sensor was located. The stability of salinity toward the new
higher values indicates the validity of this data. Further, the salinity range in open phase 2018
tested the limits of the ISFET senor which had not been tested for stability below a salinity of 9
(Gonski et al., 2018), but appeared stable here. Open phase 2019 had a narrower range of salinity





which correlated robustly with $pH_T$ as values above 8.0 were only observed when salinity was >
25. While the interdependence between $pH_T$ and salinity can be variable in nearshore systems
(Carstensen and Duarte, 2019), the degree to which $pH_T$ remained stable across a range of
salinity in open 2018 is notable. Similarly, a recent study in Stefansson Sound (~ 160 km west of
Kaktovik Lagoon) found that salinity-dependent nearshore $pH_T$ varied by year, however, the
range of salinity was more attenuated than in Kaktovik (Muth et al. 2020 *in review*). The
disparity between the salinity-$pH_T$ correlation between the open 2018 and open 2019 phases was
observable in the frequency response of variability. In open phase 2018, the PSD of $pH_T$ was low
and mostly incongruent with the frequency response of salinity. This was not the case in open
phase 2019 where the highest PSD was recorded at the same frequency (1.03 $d^{-1}$) as salinity,
which was slightly offset from the PSD peak in tidal frequency at 0.98 $d^{-1}$. These associations
suggest that events driving low salinity such as stream runoff were likely too irregular, or too
low of flux, relative to the weak but consistent tidal signal driving open ocean exchange. This
also corresponds to the lower range of salinity observed in open phase 2019 than in open phase

2018.


## 4.2  High-frequency pH in Arctic and Subarctic

Interannual variability of $pH_T$ between open phase 2018 and open phase 2019 is not dependent
on a single driving factor, including time of season. In the 2018 open phase $pH_T$ was consistently
high during a period when daylength was shortening and temperatures were falling. The
increasing trend of consistently high $pH_T$ continued into the closed phase. Conversely, August
2019 $pH_T$ had a running average that was ~ 0.2 units lower than 2018 and continued a downward
trend until the end of the time series. Similar findings have shown significantly different



interannual variability in pH along the Arctic coast that exceeded the running average difference
of ~ 0.2 observed in Kaktovik Lagoon by double (Muth et al. *in review)*. This seasonally shifting
dependence of $pH_T$ on salinity has implications for carbonate chemistry dynamics and how $pH_T$
is modified. Freshwater input from rivers have been shown to increase dissolved inorganic
carbon and lower $A_T$ which can decouple the linear relationships between calcium carbonate
saturation state, $PCO_2$, and pH (Cai, 2011; Hales et al., 2016; Salisbury et al., 2008). Glacial ice-
melt in subarctic waters, however, is unique in that its profile is low in $PCO_2$ and $A_T$ (Evans et
al., 2014). Both modes of freshwater carbonate chemistry decoupling may be present in
Kaktovik, but evidence here suggests that salinity is a non-reliable indicator of these decoupling
mechanisms as $pH_T$ values can exist across a wide range of salinity and even lack relationship
during open phases.

Open phase 2019 displayed highly variable $pH_T$ relative to open phase 2018 with an

inconsistent frequency of variability. In the subarctic waters off Alaska's south-central coast,
Jakolof Bay had a consistent seasonal trend in $pH_T$ variability with hourly rates of change as high
as 0.18 (Miller and Kelley 2020, *in review*). While these rates of hourly change are considered
high (Hofmann et al., 2011), both open phases in Kaktovik were more than double that (0.401
and 0.467) of Jakolof Bay. These extreme rates of change in Kaktovik can be partially explained
by the photosynthetic and respiratory activity within the lagoon.

**4.3   PAR and pH**
This study found robust correlations between underwater PAR and daily average $pH_T$. The
episodic nature of $pH_T$ variability in Kaktovik Lagoon was more prevalent during periods of high
underwater PAR indicative of coupled diurnal photosynthesis-respiration cycles. Consistent



levels of PAR appeared to be associated with sustained daily average $pH_T$ while drops in PAR
lowered the overall baseline $pH_T$. The rapid response of baseline $pH_T$ to PAR highlights the
tenuous balance between the biological processes that drive $pH_T$ modification. This phenomenon
is counter to what was observed in the subarctic macroalgal-dominated waters of Jakolof Bay
where the system maintained net autotrophy for a period > 60 days (Miller and Kelley, *in*
*review*). Possible explanations for the precarity of a dominant autotrophic or heterotrophic
system may be due to the shallow nature of the lagoon and frequent homogeneity of the water
column. In the shallow waters of the lagoon, high winds easily resuspend organic material,
enhance respiration, and increase light attenuation (Capuzzo et al., 2015; Moriarty et al., 2018).
Thus, small decreases in underwater PAR can lead to net heterotrophy. This supports the
sediment "food bank" hypothesis as continuous primary production is not needed to sustain
heterotrophic activity, since stored, labile, benthic OM can accumulate in shallow environments
fueling respiration (Harris et al., 2018; Mincks et al., 2005). A "bank" of OM could explain why
high levels of PAR led to a sustained $pH_T$, and any instantaneous drop in PAR was immediately
followed a decrease in daily average $pH_T$. This would suggest that high levels of PAR are only
able to offset high rates of heterotrophy which are sustained by the seasonal accumulation of
carbon subsidies from autochthonous ice algae, phytoplankton, and influx of OM from terrestrial
sources—which are likely to vary annual.

**4.4  Sea ice effects on carbonate chemistry**
A unique feature of ice covered Arctic coastal waters is the negative relationship between $pH_T$
and salinity, which was observed here and in previous studies (Fransson et al., 2013; Miller et
al., 2011; Muth et al., *in review*; Nomura et al., 2006). In the open ocean, salinity is positively



correlated with $A_T$ as higher salinity increases the difference between conservative cations to
anions. Furthermore, $A_T$ positively correlates with pH, and a higher $A_T$ is associated with a
higher buffering capacity. The formation of sea ice, however, induces cryoconcentration of DIC
via active rejection of $HCO_3^-$ during freezing and exclusion of other ions creating high salinity
brine drainage (Fransson et al., 2013; Hare et al., 2013; Miller et al., 2011). The immediate effect
of high DIC concentration can lead to the precipitation of $CaCO_3$ in the form of ikaite (a
polymorph of $CaCO_3 \cdot 6H_2O$) along the bottom of bulk ice formation generating $CO_2$ as a product
of the reaction and leading to a decrease in pH (Fransson et al., 2013; Hare et al., 2013; Rysgaard
et al., 2012). In addition, the extreme salinity and temperature in winter affect carbonate
chemistry by modulating solubility, where an increase in salinity decreases $CO_2$ solubility, and
colder temperatures increase $CO_2$ solubility. These salinity and temperature conditions result in a
volatile thermodynamic stability of $CO_2$ where salinity effects outweigh temperature effects and
can facilitate a degassing of $CO_2$ (Papadimitriou et al., 2004).

The continually decreasing $pH_T$ observed in this study suggests that these carbon

concentrating corollaries of sea ice formation may be in effect and contribute to the negative
relationship observed between $pH_T$ and salinity. That is, if there is no outgassing of $CO_2$, the
relative increase in DIC and concomitant decrease in pH will be equal to that of salinity. During
ice coverage, the running average of $pH_T$ decreased from 7.93 in the beginning of November, to
7.56 in late April, and mirrors the under-ice salinity trend. This decrease is nearly identical to the
0.4 pH drop observed in the upper 2 m below the ice in Amundson Gulf from the November to
April period (Fransson et al., 2013). While this phenomenon could explain the general
decreasing trend between $pH_T$ and salinity, it would be remiss to state that this negative
correlation is entirely driven by cryoconcentration and ikaite formation. What is more likely is





that cryocentration is occurring in tangent with accumulated aerobic respiration byproducts
overtime, and the high frequency of $pH_T$ variability is the result of biological and thermodynamic
processes on carbonate chemistry.

**4.5   Under ice variability in pH**
The frequency of $pH_T$ variability under ice cover was inconsistent. The PSD was weak overall
during the closed phase but had a peak at 0.39 $d^{-1}$, which corresponded to a peak in temperature
around the same frequency 0.36 $d^{-1}$. The temperature range of 1.9 ˚C during the closed phase can
affect carbonate chemistry thermodynamics potential modulating pH by ~ 0.036; however, this is
less than the derived $pH_T$ uncertainty. The other factor driving $pH_T$ variability is biological
respiration. Data sonde measurements of dissolved oxygen recorded in late April showed bottom
waters reaching lows of 5.0 mg $L^{-1}$ (43 % saturation) compared to surface levels of 11.5 mg $L^{-1}$
(94 % saturation) (Table S1). The stratification of oxygen in this case can likely be associated
with burgeoning PAR levels in April. Previous studies have shown increases in pH are
associated with photosynthesis during ice-cover, which is more prevalent proximal to bulk ice
resulting in higher pH at the surface compared to the bottom (Matson et al., 2014). Other factors
driving pH variability could be due to the competition between anaerobic and aerobic
metabolism in low oxygenated water, and the transfer of reduced metabolites from bioirrigation
(Aller, 1982, 2001; Zakem et al., 2020). Efflux of reduced metabolites from the sediment can
lead to high concentrations of reduced inorganic nitrogen if oxygen concentrations are low and
oxidation processes slow (Aller, 2001; Middelburg and Levin, 2009). Discrete samples taken in
April found high concentrations of reduced nitrogen in the bottom waters (Table S1). If oxygen
levels begin to increase in late spring due to photosynthesis, the subsequent oxidation of nitrogen



and other accumulated reduced metabolites could decrease pH as was seen from mid-April to
mid-May. Due to limited under-ice sampling, however, there is no way to determine the
trajectory of oxygen decrease or exact timing of under ice photosynthesis. The only other
mechanism potentially supplying oxygen to the lagoon would be associated with water mass
exchange via tide. According to the frequency analysis, there is limited evidence showing a
correlated frequency peak between $pH_T$ and tide, indicating that tidal exchange may be restricted
or not a modulator of $pH_T$ during the closed phase. Without measuring dissolved oxygen,
however, it remains unclear if oxygen is the determinant factor driving $pH_T$ modification during
the closed phase.

### 4.6 Arctic lagoons as carbon source to atmosphere

The estimates of $CO_2$ flux during the open phases of 2018 and 2019 were an *a posteriori* method
to examine the drivers of pH variability in Kaktovik Lagoon. Following this approach,
comparisons between $pH_T$ rate of change and estimated $CO_2$ flux did not correlate, suggesting
that outgassing rates were not significant enough to raise *in situ* pH. Rather, the analysis showed
that the estimated lagoon $CO_2$ flux varied substantially by year and appears at times to be a
source of $CO_2$ to the atmosphere. This is counter to other studies that measured carbon flux at a
lagoon in the far western Beaufort (Elson Lagoon), where this site was categorized as a carbon
sink; however, these lagoons differ in size, residence time, and connectivity to adjacent water
bodies (Lougheed et al., 2020). Overall, the western Arctic Ocean is thought to be a carbon sink
(Evans et al., 2015a; Laruelle et al., 2014); although Mathis et al. (2012) described occasional
storm-induced upwelling events across the Beaufort Sea shelf that cause $CO_2$ efflux to the
atmosphere. In this study, the variability in estimated flux from the lagoon appeared to be a





function of baseline $pH_T$ more than wind driven stress. Open phase 2018 had a higher baseline
$pH_T$ (8.01 – 8.18) than open phase 2019 (8.04 – 7.72), and despite wind speeds comparable to
open phase 2019, resulted in less estimated $CO_2$ efflux to the atmosphere. Conversely, open
phase 2019 maintained a lower baseline $pH_T$ which promoted favorable disequilibrium (i.e.,
difference between $PCO_{2sw}$ and $PCO_{2a}$) conditions that only needed wind stress as a catalyst.
Since flux preceded low $pH_T$ values, and outgassing did not decrease hourly $pH_T$, the
mechanisms driving low pH and $PCO_2$—likely biological respiration— transcend the
counterbalance of outgassing.

The flux estimates in this study suggest that the novel characteristics of coastal lagoons

should be considered anomalous compared to the greater across shelf Arctic coast, defined as
waters north of 70 ˚N and 100 ˚W (Bakker et al., 2014). The current classification of the coastal
Arctic does not account for lagoons as specific ecosystems. Thus, the western Arctic coastal
ocean is defined as a relatively homogenous area $1.2 \times 10^{12}$ $m^2$ along the Chukchi and Beaufort
Seas extending 400 km offshore (Evans et al., 2015a). The coastal Beaufort Sea under this
definition is estimated to have an annual mean carbon uptake of 8.5 Tg C $yr^{-1}$ without ice, and a
daily annual mean flux of -2.1 mmol $CO_2$ $m^{-2}$ $d^{-1}$ (Evans et al., 2015a). Recent evidence,
however, has shown that previous estimates of the carbon sink capacity of the Arctic Ocean have
been overestimated, suggesting that current and increasing riverine discharge will cause a
reduction in $A_T$ ultimately decreasing its potential to absorb $CO_2$ (Woosley and Millero, 2020).
While the lagoon ecosystems comprise a small proportion of the greater Beaufort Sea shelf, they
encompass > 50 % of its coastline with significant freshwater inputs that can lower the carbon
sink capacity (Dunton et al., 2006; Woosley and Millero, 2020). It is suggested here that certain
lagoons, including Kaktovik, are likely episodic sources of $CO_2$ to the atmosphere during open



phases. The daily average ($\pm$ s.d.) $CO_2$ flux for Kaktovik Lagoon during open phase 2018 and
2019 was -2.2 $\pm$ 6.5 and 14.6 $\pm$ 23.9 mmol $CO_2$ m$^{-2}$ d$^{-1}$, respectively. Over the entire calendar
year that encompasses both open phases during which sensors were deployed, the annual daily
average flux was 5.9 $\pm$ 19.3 mmol $CO_2$ m$^{-2}$ d$^{-1}$ for the entire calendar year. If integrated over the
entire open phase (51.58 d in 2018 and 49.38 d in 2019), and the area of Kaktovik Lagoon,
estimates suggest a net carbon flux of -2.68 x 10$^{-5}$ Tg C open$_{18}$$^{-1}$ in open 2018 and 1.67 x 10$^{-4}$ Tg
C open$_{19}$$^{-1}$ in open 2019. It is noted that these estimates are for incomplete open phases as the
data presented here do not comprise the entirety of each seasons due the scheduling of SeaFET
deployment and recovery. If incorporating all the lagoons along the coast, it is plausible that the
source of $CO_2$ from the lagoon ecosystems would partially offset the carbon sink capacity
previously established, particularly when considering that the estimated daily annual average
flux is at times substantially greater (5.9 $\pm$ 19.3 mmol $CO_2$ m$^{-2}$ d$^{-1}$), and opposite, of current
estimates (-2.1 mmol $CO_2$ m$^{-2}$ d$^{-1}$) (Evans et al., 2015a; Mathis et al., 2015). Further studies that
can capture high-frequency carbonate chemistry variability are needed though to determine the
degree and frequency of the Beaufort lagoon ecosystems' air-sea carbon exchange.

There is a fair amount of confidence in these estimates because the $A_T$-salinity correlation

was robust ($R^2$ = 0.968) and the regression coefficients were proximal to other $A_T$-salinity
regressions for the Gulf of Alaska and the western coastal Arctic, despite being derived from
only three discrete samples (Evans et al., 2015b; Shadwick et al., 2011; Yamamoto-Kawai et al.,
2005). Further, the overall uncertainty of the flux estimates was low. The main source of
deviation was associated with higher $PCO_2$ values calculated from the $A_T$-salinity$_{in\ situ}$ regression.
This made up the upper bound of uncertainty, thus, the conclusions drawn here are from the
more conservative flux estimates. The effect of fresh water on the gas transfer velocity





comprised the lower bound of the uncertainty and was negligible overall. For the flux estimates
presented here, a homogenous water column with respect to pH was assumed, given that discrete
sonde measurements only showed pH stratification during the ice-covered season (Table S1).
This is not to suggest that salinity and temperature driven stratification do not exist, rather that
the evidence here suggests $pH_T$ water column homogeneity. For example, $pH_T$ during open phase
2018 did not correlate with salinity as values > 8.01 were present across a salinity range of 25. In
cases where $pH_T$ positively correlated with salinity as seen during open phase 2019, a freshwater
stratification would suggest that low salinity at the surface would be associated with lower $pH_T$,
and likely increase $CO_2$ flux as there would be a greater disequilibrium between the lagoon and
the air. According to the quadratic fit between $pH_T$ and salinity, lower $pH_T$ at the surface
associated with freshwater stratification would outweigh the $A_T$ estimates based on salinity by an
order of magnitude if there was a salinity difference of 10 between the surface and bottom
waters. Thus, freshwater stratification at the surface would likely exceed our upper bound flux
uncertainty and increase efflux rates. Further, any modulation of flux by temperature on the gas
transfer velocity are less than the estimated upper bound uncertainty and considered negligible.

**5   Conclusions**
This study presents the first high-frequency pH time series for the open and under ice phases in
the coastal Arctic lagoon system. Uncertainty estimates for $pH_T$ were higher than desired but
describe general trends and relative rates of change that are informative for understanding pH
variability. The extremely low anomaly between the reference $pH_T$ sample and the SeaFET
suggest that the uncertainty is likely lower than estimated. pH can vary dramatically by year for
the open phases and is likely a function of PAR availability and the amount of OM delivered





from terrestrial sources as the balance between system autotrophy and heterotrophy were
tenuous. This resulted in hourly $pH_T$ rates of change > 0.4 units. Under ice pH variability
exhibited complexities, and we postulate that multiple drivers of pH variability such as carbonate
chemistry thermodynamics, ikaite precipitation, and sediment efflux were all contributing
mechanisms. It is apparent that further studies of carbonate chemistry dynamics at the sediment-
water interface are needed to help elucidate porewater effects on bottom water pH variability
during the ice cover phase, as well as continuous oxygen measurements. Estimated $CO_2$
outgassing during the open phase was not a significant factor driving $pH_T$ variability due to the
collinearity of wind stress and the infrequent convergence between disequilibrium and wind
speed. However, carbon flux estimates suggest that the Beaufort lagoon ecosystems may be a
substantial source of carbon to the atmosphere, which is counter to previous studies predicting
coastal Arctic waters as a $CO_2$ sink. These results highlight the need for further investigation of
the Beaufort lagoon ecosystems in the context of carbonate chemistry dynamics, as these
processes can affect the diverse biological communities that are present here, and aid in
understanding western coastal Arctic biogeochemical dynamics.

**Data availability**: All data accessed from the Beaufort Lagoon Ecosystems LTER is available
on the Environmental Data initiative. See reference section for access links.

**Author Contributions**: Cale A. Miller, NM, CB, and ALK conceptualized the manuscript
thesis. CAM performed the data analysis and data visualization. ALK performed initial data
QA/QC for pH data. ALK, NM, and CB performed lab analyses. Cale A. Miller performed all
data analysis. CAM wrote the original manuscript draft with minor contributions in the



introduction from ALK and CB in the methods. ALK, CB, and NM reviewed and edited the
manuscript.

**Competing interests**: The authors declare no conflict of interest.

**Acknowledgments**: We thank R/V Proteus captains Ted Dunton and John Dunton for expert
mooring deployment and recovery. We additionally thank K. Dunton, S. Jump, J. Kasper for
logistical and field assistance. This work took place in the traditional and current homeland of
the Kaktovikmuit.

**Financial support**: This material is based upon work supported by the National Science
Foundation under award #1656026

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






**Table 1**. Calibration and reference bottle data for SeaFET. Propagated uncertainty, for each
bottle, and the calculated total pH uncertainty value as overall average (in bold).

| Date & Time | Source | $pH_T$ internal electrode | Propagated uncertainty | Anomaly: \| bottle sample - SeaFET \| |
|---|---|---|---|---|
| 17 Aug. 2018 | SeaFET | 8.076 | — | |
| | Bottle sample | 8.073* | 0.1600 | — |
| 26 Apr. 2018 | SeaFET | 7.576 | — | |
| | Bottle sample | 7.582 | 0.1006 | 0.0061 |
| **Total uncertainty** | | | | **0.0889** |







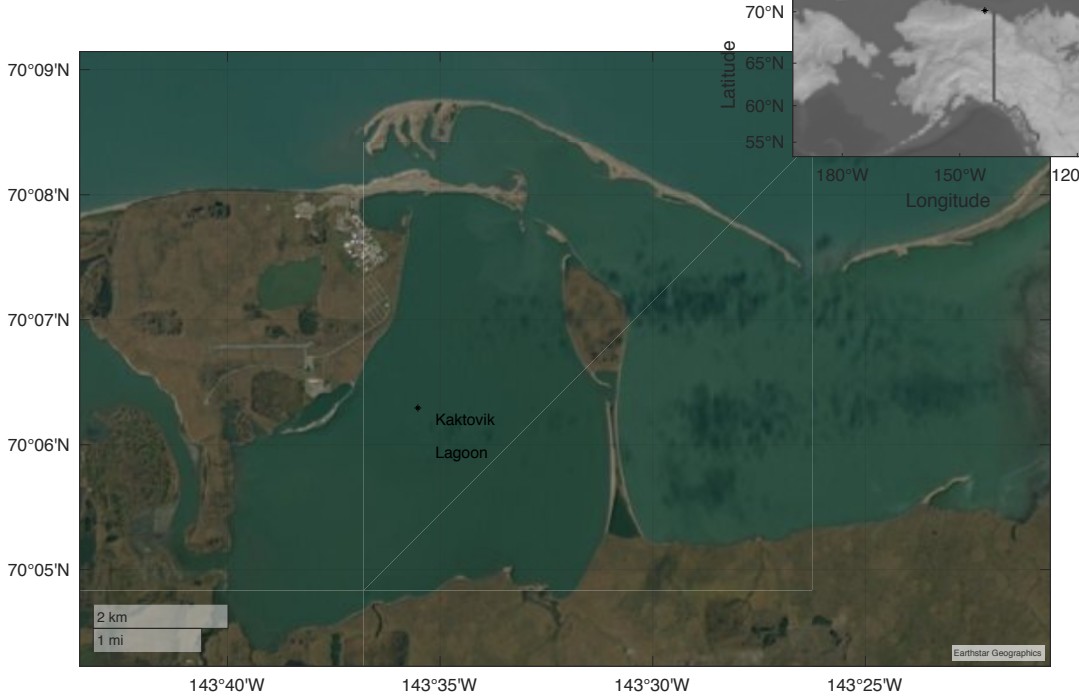


**Figure 1**. Study site at Kaktovik Lagoon along the Beaufort Sea Coastline.






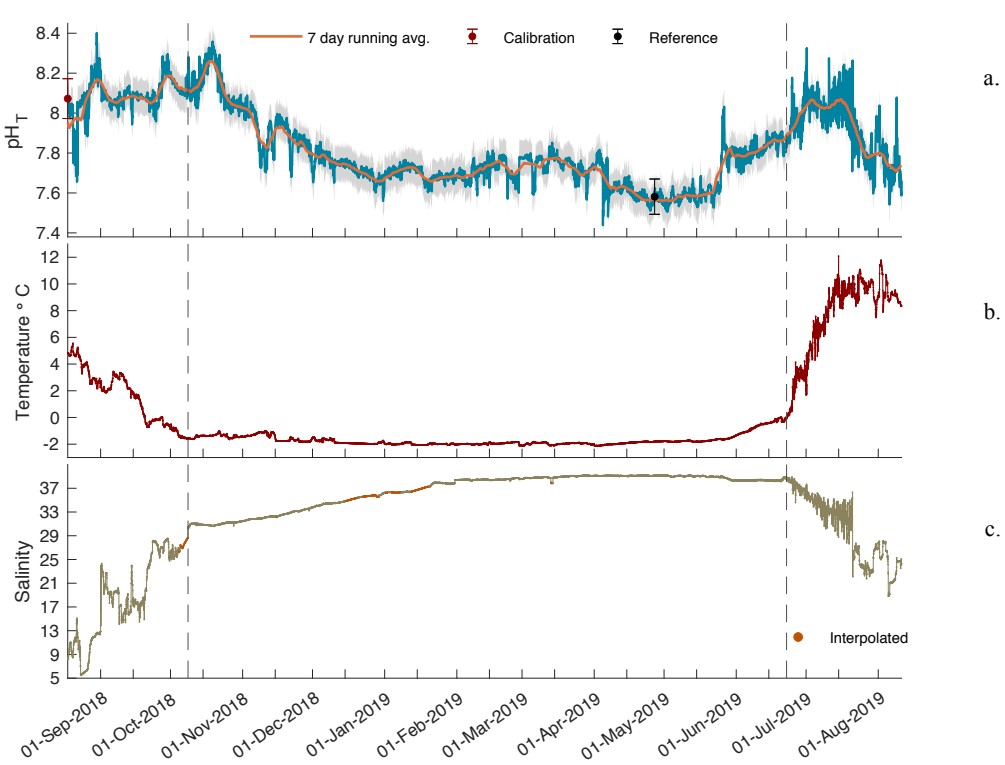

**Figure 2.** Times series of pH$_T$ (a), temperature (b), and salinity (c) in Kaktovik Lagoon for entire

deployment period from 17 August 2018 to 11 August 2019. The first section to the left of the

dashed line is open phase 2018, the middle section is closed 2018 – 2019, and the last section to

the right of the second dashed line is open phase 2019.

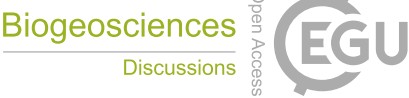


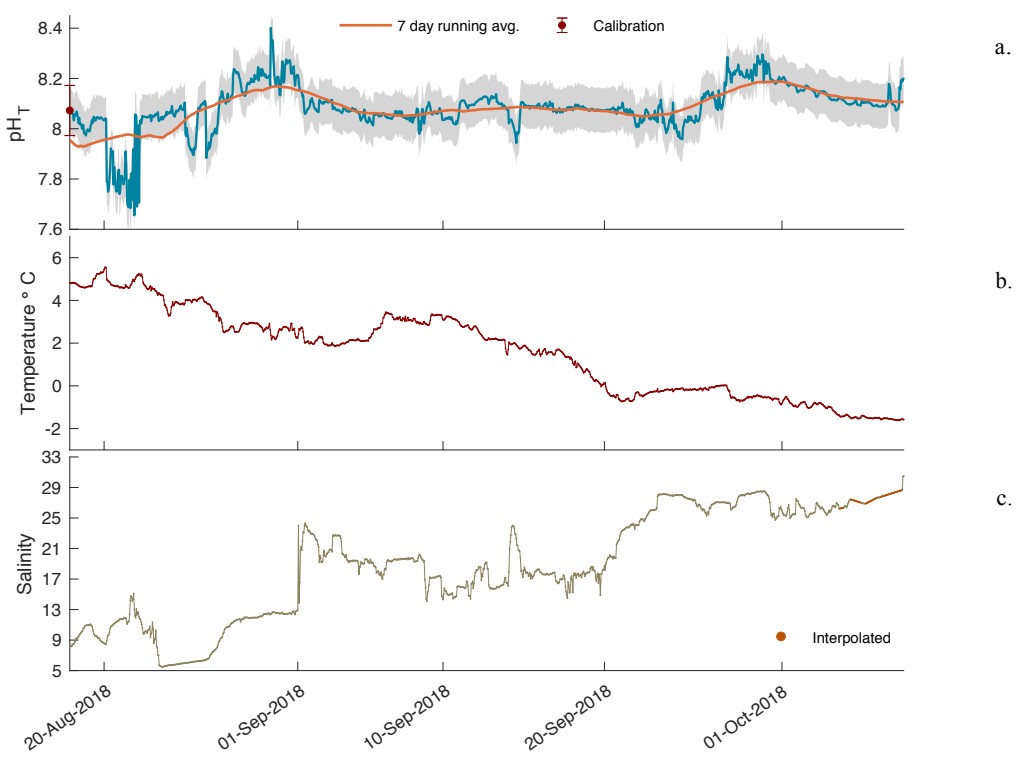

**Figure 3.** Open phase 2018 time series of pH$_T$ (a), temperature (b), and salinity (c) in Kaktovik
Lagoon.






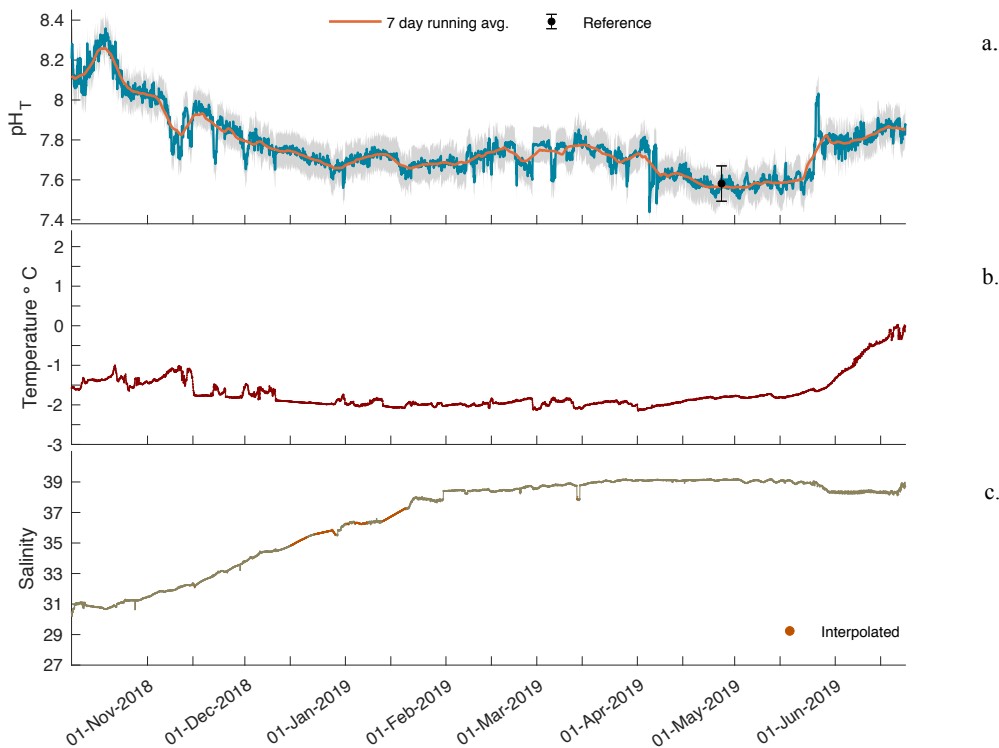

**Figure 4.** Iced phase 2018 – 2019 time series of pH$_T$ (a), temperature (b), and salinity (c) in
Kaktovik Lagoon.






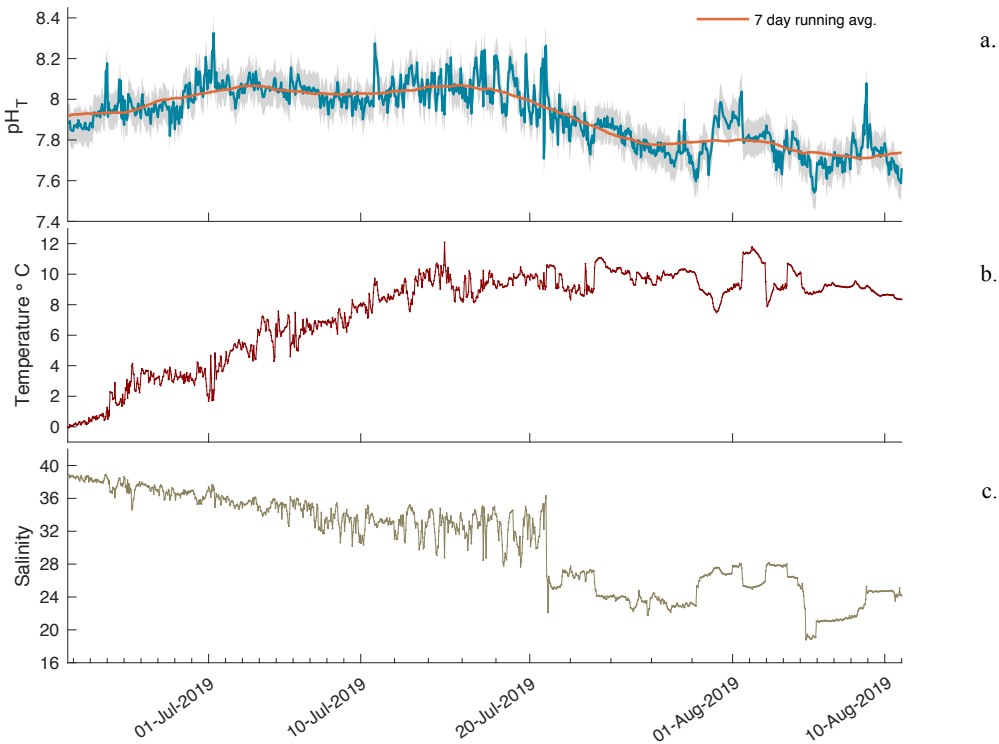

**Figure 5.** Open phase 2019 time series of pH$_T$ (a), temperature (b), and salinity (c) in Kaktovik
Lagoon.



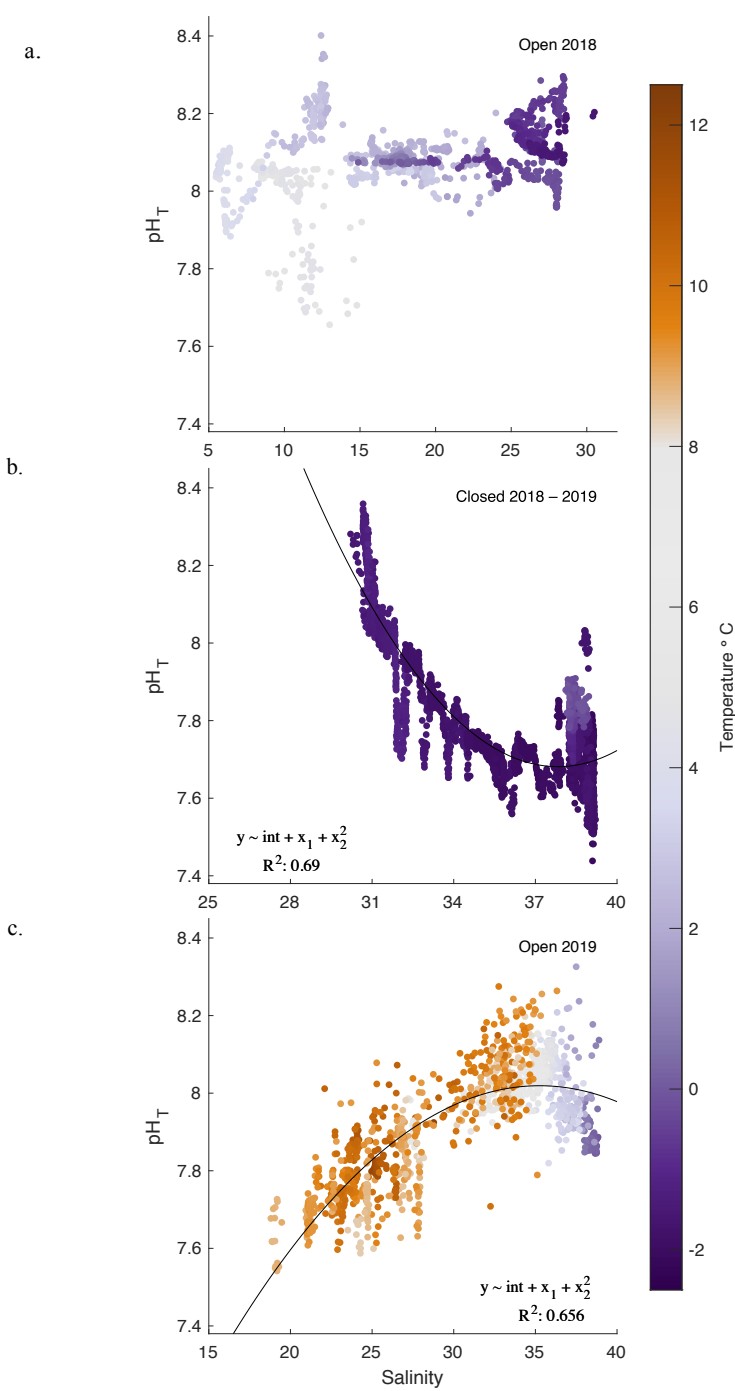

**Figure 6**. pH$_T$-salinity correlations for open 2018 (**a**), iced 2018 – 2019 (**b**), and open 2019 (**c**). Quadratic fits are applied to iced and open 2019 phases only. Temperature is represented in color for all correlations.







**Figure 7**. Power Spectral Density (PSD) plots for $pH_T$ (a,b,c), temperature (d,e,f), and salinity (g,h,i) at each phase of the time series: open 2018 (top row), ice-cover 2018 – 2019 (middle row), and open 2019 (bottom row).






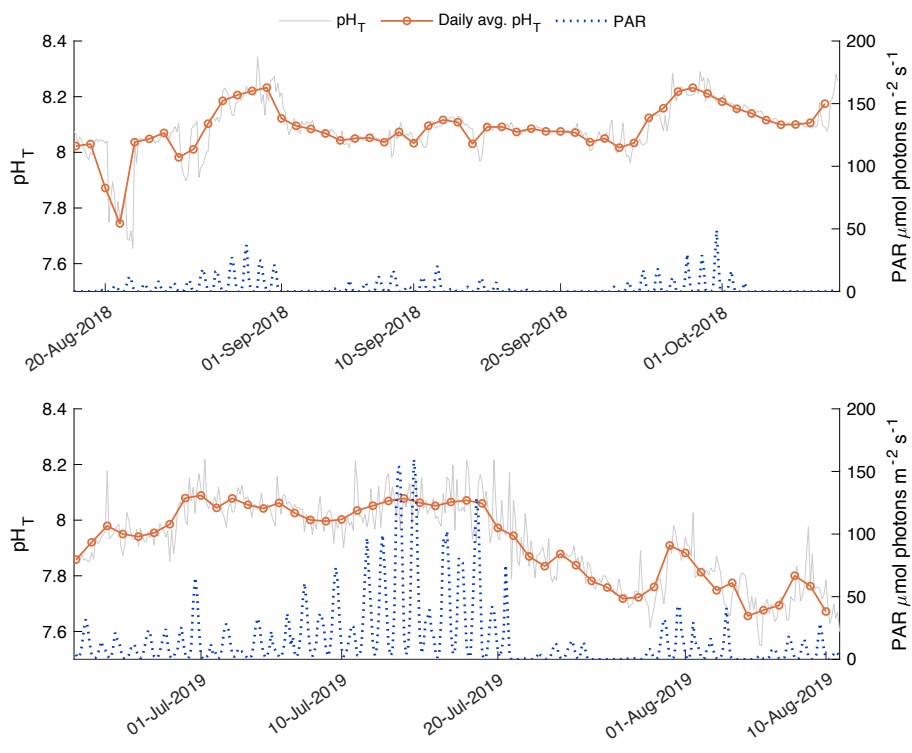

**Figure 8**. $pH_T$ (gray line) and PAR (blue dots) for open phase 2018 (**a**) and open phase 2019 (**b**). Daily
average $pH_T$ (orange line) is displayed overtop hourly variability.





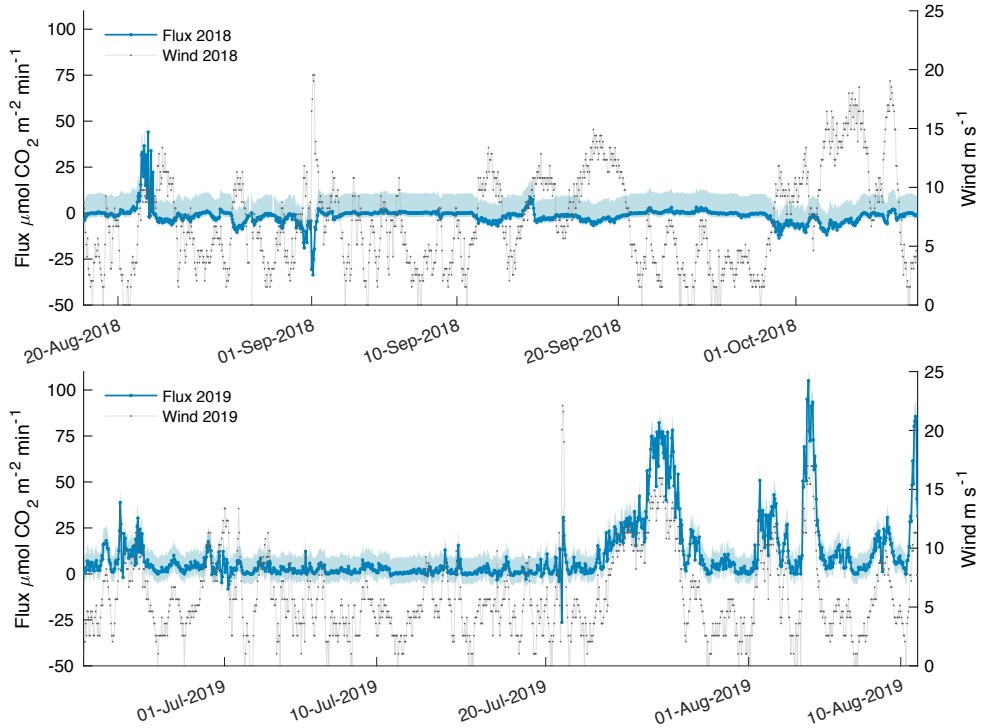

**Figure 9**. Estimated carbon flux (orange) and wind speed (grey) for open phase 2018 (**a**) and open
phase 2019 (**b**). Uncertainty around each estimate is shaded in blue where the upper bound is
associated with difference in $PCO_2$ from the $A_T$-salinity$_{in\ situ}$ regression, and the lower bound
associated with freshwater Schimdt number. The upper and lower bounds for
open 2018 were 10.67 and 2.23 μmol C m$^{-2}$ min$^{-1}$ while open 2019 upper and lower bounds were 8.56
and 5.52 μmol C m$^{-2}$ min$^{-1}$, respectively.