# Peer review of "Cale A. Miller1,3, Christina Bonsell2, Nathan D. McTigue2, Amanda L. Kelley3"

_Biogeosciences, 2020_

## Referee Comment (RC1) · Anonymous Referee #1 · 4 Nov 2020

Comments to Author:

1 Overview

In this study, Miller et al. describe an entire calendar year of pH, temperature and salinity changes in the Kaktovik Lagoon, Alaska. Using time series of current speed and photosynthetically active radiation, they look for the mechanisms driving these pH changes. They then go further and, using wind speed and atmospheric $pCO_2$ measurements, provide estimates of $CO_2$ fluxes to the atmosphere in the ice-free period. The study reads very well, is scientifically interesting, novel, and should be published given addressing a few points detailed below. .

2 Major points

a. The pH dependency on salinity and temperature could be quantified more rigorously than with the present regression. As is, it is unclear how exactly this regression was performed, why it was done like that, etc. The approach of Hagens and Middelburg (2016), who wrote a paper on how to attribute pH variability to governing factors, seems more robust and I would encourage the authors to look in this direction. Looking at Fig. 2, it is hard to believe that salinity and pH are not correlated in the first open phase, as stated on line 372. What was the rationale for applying a 7-day running average on the pH time series? Why not doing it also on the salinity and temperature time series and look for correlations in the smoothed time series instead of in the noisy ones?

b. The authors should go further in the interpretation of the CO2 flux estimates. Clearly, PAR is not correlated with CO2 flux estimates, and something else than biology must govern the high CO2 flux variability. Looking at Fig. 9, it seems that changes in CO2 fluxes are caused by storms and other weather events, rather than biological processes. What about adding current speed (from supp. Fig. 1) on Fig. 9? It would give a better idea whether weather events induce are translated into water turbulence or not. In a shallow lagoon (4.4 meters max) such as that, storms would likely resuspend sediments, releasing DIC in the bottom waters and lead to short term CO2 efflux from the lagoon to the atmosphere. Measuring transmissivity would have helped to quantify sediment resuspension through time. The differences between the CO2 flux and the pH time series, in terms of behavior and possible controlling mechanisms, should be emphasized more in the discussion, abstract and possibly title.

c. Doing the integral over time of the CO2 flux to/from the atmosphere would allow to put a nice number on the source or sink behavior of this lagoon. Given that, another point to consider for discussion would be: as the ice extent drops in the next century due to temperature increase, how could that affect CO2 air-water fluxes in these lagoons?

d. The whole frequency analysis and its results were quite unclear to me. Specifically, it would be very helpful to indicate or highlight on Fig. 7 the range of frequencies that

are representative of given natural events (e.g., tides, seasons, etc.). .

3 Minor points

Title: Consider something more meaningful/informative than "non-linear extremes"

L. 56: Define LTER

L.59: Here and elsewhere: because pH is on a log scale, it is a bit meaningless to simply give the magnitude of a pH change (see the technical note from Fassbender et al., 2020, published recently as a preprint in Biogeosciences). For instance, instead of saying "pH varied with a difference of 0.2 units" without giving the actual pH value before, say "pH varied from 7.8 to 8".

L. 65: "Efflux" is unclear, specify in which direction it's going. Given the large error bar, we can't actually say that it's going in any particular direction. . .

L. 66: Explain what are the "geomorphic differences"

L. 105: Specify "specific water mass mixing patterns"

L. 108: Acidification results in "lower" pH and saturation state, but not necessarily "low"

L. 110: Here and in the next sentence: which calcium carbonate mineral are you talking about?

L. 118: Here and elsewhere: writing pCO2 with a capital P is strange, it is usually written "pCO2"

L. 146: "Meteorological events" is too vague

L. 186: Can you show the Arey and Jago lagoons on Fig. 1?

L. 207: Were sediments also retrieved in April, June and August?

L. 235: About using the Lueker et al. constants: Dinauer and Mucci (2017) showed that carbonic acid dissociation constants (K1* and K2*) of Cai and Wang (1998) seem

to be more adapted to low-salinity environments such as this lagoon. Papadimitriou et al. (2018), Millero et al. (2002) and Sulpis et al. (2020) all obtained K1* and K2* values that are lower than those from Lueker et al. (2000) in cold waters such as those from the present lagoon. Each of these estimates, including those from Lueker et al., come with associated uncertainties, but what is rarely taken into account is the uncertainty associated with the choice of constants that one choses for an analysis, because it's very hard to quantify. This additional "hidden" source of uncertainty should be discussed here, especially given how extreme this lagoon is in terms of temperature and salinity.

L. 279: What is a "sidelobe attenuation"?

L. 290: I didn't get "Measurements identified as below the freezing point of water"

L. 305: Can you show the Barrow and Barter Island airport on the map in Fig. 1?

Eq. 2: what are the brackets around U2 for?

End of section 2.5: I didn't understand what the upper/lower bound uncertainties are and what is the link with Eq. 1

Fig. 6: replace y and x by the actual variable names. Are x1 and x2 both salinity?

In the end, what were the porewater measurements for? .

4 References

Cai, W.J., Wang, Y., 1998. The chemistry, fluxes, and sources of carbon dioxide in the estuarine waters of the Satilla and Altamaha Rivers, Georgia. Limnology and Oceanography 43, 657-668.

Dinauer, A., Mucci, A., 2017. Spatial variability in surface-water pCO2 and gas exchange in the world's largest semi-enclosed estuarine system: St. Lawrence Estuary (Canada). Biogeosciences 14, 3221-3237.

Hagens, M., Middelburg, J.J., 2016. Attributing seasonal pH variability in surface ocean waters to governing factors: Governing factors of seasonality in pH. Geophysical Research Letters 43, 12528-12537.

Lueker, T.J., Dickson, A.G., Keeling, C.D., 2000. Ocean pCO2 calculated from dissolved inorganic carbon, alkalinity, and equations for K1 and K2: validation based on laboratory measurements of CO2 in gas and seawater at equilibrium. Marine Chemistry 70, 105-119.

Millero, F.J., Pierrot, D., Lee, K., Wanninkhof, R., Feely, R., Sabine, C.L., Key, R.M., Takahashi, T., 2002. Dissociation constants for carbonic acid determined from field measurements. Deep Sea Research I 49, 1705-1723.

Papadimitriou, S., Loucaides, S., Rérolle, V.M.C., Kennedy, P., Achterberg, E.P., Dickson, A.G., Mowlem, M., Kennedy, H., 2018. The stoichiometric dissociation constants of carbonic acid in seawater brines from 298 to 267 K. Geochimica et Cosmochimica Acta 220, 55-70.

Sulpis, O., Lauvset, S.K., Hagens, M., 2020. Current estimates of K1*and K2* appear inconsistent with measured CO2 system parameters in cold oceanic regions. Ocean Sci 16, 847-862.

---

## Short Comment (SC1) · 23 Nov 2020

Stephen Gonski

sgonski26@gmail.com

Lines 487-489 - The lowest salinity of the sensor data (pH_INT and pH_EXT) presented in Figure 3c of Gonski et al. (2018) was S=3.26 while the lowest salinity sensor data with coincident bottle samples collected with which to compare was S∼9. pH_INT was stable at both places while pH_EXT was not at the former. A follow-up paper to Gonski et al. (2018) is currently is in the final stages of preparation that looks at the performance of the Durafet and its internal and external electrodes down to S<0.5 at sub-zero temperatures in a dynamic euryhaline estuary as well. The authors should clarify how "stability" down S∼5 was verified (i.e., with sensor data only or sensor data with coincident bottle sample data) since the sentence in its current form is inaccurate.

[Figure]

Lines 1024-1026 - Further salinities of validation samples should be added to Table 1 for clarity if language about salinity will be kept as well. Finally the date of collection for the second validation sample should be changed to 26 Apr. 2019 since 26 Apr. 2018 is not covered by the data presented in Figure 2.

Other than that, congratulations on an excellent manuscript and best of luck seeing it through to publication.

Please also note the supplement to this comment:
https://bg.copernicus.org/preprints/bg-2020-358/bg-2020-358-SC1-supplement.pdf

**Supplement:**

[supplement omitted: unrelated document]

---

## Short Comment (SC2) · 23 Nov 2020

We thank Stephen Gonski for the details and comment regarding salinity and will promptly make the corrections pointed out here in the revised manuscript.

---

## Referee Comment (RC2) · Anonymous Referee #2 · 1 Dec 2020

The authors present the results of annual pH measurements in an Arctic lagoon. They found that salinity-pH relationship differs between observations periods, and that the lagoon acts as $CO_2$ source to the atmosphere during the open season. These observations and findings are interesting and valuable. I recommend that this paper be published in Biogeosciences with minor revisions.

Specific comments:

1. AT is estimated from S- AT regression line based on 3 samples and is used to calculate $CO_2$ flux. Possible errors in $CO_2$ flux are estimated by comparing estimations using Ssensor-AT and Sbottle-AT. However, S change due to mixing with seawater, mixing with river water, melting and formation of sea ice can change S-AT relationship. This is of particular concern during the melting season (open 2019) when no AT sam-
pling was made. Also, high concentration of organic matter in some coastal regions can significantly contribute to AT. This may be important for the lagoon system. These uncertainties should be discussed in the text.

2. Also about the error in CO2 flux. I do not understand why uncertainty associate with freshwater Sc and AT-S regression should represent lower and upper bound? How about additive errors?

3. Different pH-S relationship between two open periods is interesting but not clearly explained in the text. Open 2018 is in the fall, when vertical mixing can be enhanced by cooling and storms. Accordingly, air-sea CO2 exchange can result in relatively stable pH (and pCO2) despite the change in salinity. On the other hand, open 2019 is in melting season when stratification, warming, dilution, and remineralization at the bottom can largely change pH. Can this explain the difference?

4. I suggest authors to add a figure showing estimated pCO2 time series. It would help to better understand seasonal variability in carbonate chemistry.

Minor points:

Figure 1: The station location and text in the picture are hard to see. What about changing to white? Please show locations of Arey and Jago lagoons and Beaufort Sea.

Figures 4, 6, 7: Change "iced" and "ice-cover" to "closed" to be consistent with the main text.

Figure 7: It is hard to compare position of peaks among figures. Adding vertical lines for some important frequencies should be helpful (1 and 2 d-1?).

Figure 8: State "detrented pH" in the caption.

Table 1 and S1: Explain what * means.

Table S1: Add in situ salinity used for AT-Sin situ regression in the table.

Figure S2: May be better to show original invalid data. Otherwise, this is exactly the same as Figure 2c.

L202 "April, August and June" : Change to "August 2018, April 2019, June 2019, and August 2019"

L203: Specify "physicochemical parameters"

Section 2.3 title: "Seawater chemistry and pH sensor calibration"

L230: "All salinity measurements were . . .." Does this mean "Salinity of all AT/pH samples were..."?

L234: Which temperature was used?

L300: Superscript "2" of "R2".

L318: (Wanninkhof, 2014) should be Wanninkhof (2014).

L329-330: I could not understand how pH was detrended for open 2018 period when there was no correlation between pH and S. This may be due to my misunderstanding. Please explain how "detrended pH" was obtained in more detail.

Section 2.6 "statistical applications": With this generic title, it is hard to understand what this section describes about. Information in this section would be better moved to the section 3.3, when this application is needed. In that case, first two sentences can be removed as they are already explained in the section 3.1.

L408: p-value <0.000 should be <0.001. It cannot be negative.

L482: It is said that "in the open phase of 2018, pH values were observed to be >8.05 despite the striking range of S from 5 to 30." However in figure 2, pH varies from <7.7 to 8.4 in open 2018 period.

L561-563: Please modify the text to state that cryoconcentration of DIC (and AT) itself lowers pH. Readers may misunderstand that cryoconcentration of DIC lowers pH only

via ikaite formation.

L572-582: Some quantitative analysis is needed. According to my test calculation using CO2sys, cryoconcentration of DIC, AT, S cannot explain observed large decrease in pH. From very high nutrient concentration in closed season, remineralization seems to have the largest impact on pH decrease. This effect can be roughly quantified from increase in nutrients with RFR. If possible, please add some quantitative analysis about ikaite formation.

L600-606: Is the winter oxygen concentration of 5 mg/L ($\sim$160 umol/L) low enough to limit nitrification?
* * *

---

## Author Comment (AC1) · 22 Dec 2020

We would like to thank both anonymous reviewers for their comments and time to help improve this manuscript.

a. We thank the reviewer for this comment and agree there is a utility in scrutinizing the pH-salinity relationship more vigorously. The approach followed by Hagens and Middleburg 2016 derives pH sensitivities as affected by acid-base constituents along with their dependence of speciation on temperature, salinity, and pressure. The relationship of pH to temperature and salinity in this respect would be due to the modulation of dissociation constants and, thus, carbonate chemistry speciation if we assume this to be the dominant acid-base dynamic in the lagoon. Hofmann et al. 2009 examined the sen-

sitivity of dissociation constants of acid-base speciation to the concomitant effects on proton concentration. They showed that the main driver of pH was the ratio of DIC:AT and a function of conservative mixing. Secondary was biological activity. In this paper, we examined the relationship between pH and temperature as a linear-quadratic. Which is an appropriate approach given the data we have collected.

As mentioned above, quantifying the effects of temperature and salinity on the dissociation constants of the carbonate system and how this modifies pH would be based on using AT data estimated from our AT-salinity regression, thus resulting in a circular approach at determining pH sensitivity to temperature and salinity. For this reason—and as suggested by the reviewer— we find the approach of examining the relationships between running average data more appropriate. Further examination we feel is beyond the scope of this manuscript given the other findings and observations presented.

We have amended the manuscript to reflect our use of this linear relationship approach between pH and salinity and temperature. See lines 349–351, 397–399, 405–406 in the revised manuscript.

We have revised the manuscript to reflect the new findings and make clear that no linear relationships exist with temperature according to our analyses, nor did we find a relationship between salinity and pH during open phase 2018.

Below is the relationship between the 7-d running average for salinity and pH during open phase 2018

We encourage the reviewer to examine figure 3 rather than figure 2 when interpreting the relationships between salinity and pH as figure 3 is a magnified perspective of this data. From this viewpoint, it is more clear that the variability between salinity and pH is non-congruent.

We choose a 7-day running average of the time series because this appeared to give a smoothed line trend that still highlighted small shifts to new baseline pH levels.

New relationships between salinity, temperature, and pH were examined and 7-day running averages were applied to salinity and temperature as well for the times series figures.

b. We find the reviewer's suggestion beneficial and agree that we can discuss the potential mechanisms of $CO_2$ flux variability to a greater depth. We note though that since we, unfortunately, do not have additional data such as transmissivity (which would be beneficial), most of our conclusions are speculative.

We had not thought to correlate flux with current speed, however, when reexamined we found no correlation between the two and feel it would not add more to the current figure as is.

We found supporting evidence in the literature regarding the reviewer's comment re upwelled $CO_2$. We have incorporated these points into the discussion. See lines 676—683 in revised manuscript. Briefly, we highlight that previous studies (Åberg et al. 2010) have found that storm events create an upwelling of $CO_2$ and erosion of stratified layers and conclude that upwelled $CO_2$ to the surface layers is greater than the $CO_2$ loss to the atmosphere. These findings support our conclusions and we feel have improved our conclusions regarding the disequilibrium being the driving factor of $CO_2$ off-gassing.

Given the already lengthy abstract, we feel that additional commentary on $CO_2$ flux would not entice readers. However, we changed the title to reflect our conclusions regarding $CO_2$ flux.

c. We have added this additional value of integrated C flux over the entire calendar year which provides a better metric for comparison to other published literature. This value is $1.4 \times 10^{-4}$ Tg C $yr^{-1}$. In addition, we have added the suggestion by the reviewer to include the potential implications this finding may have as ice-free days increases in the coming decades. See line 705 in the revised manuscript.

d. We have added additional detail regarding the frequencies associated with daily irradiance and tides. See lines 299–300 in revised manuscript.

Title: We have changed the title to "The Seasonal Phases of an Arctic Lagoon Reveal the Discontinuities of pH Variability and CO2 flux at the Air-sea Interface"

L. 56: We have defined LTER

L. 59: We have amended this sentence and those similar that only reported a magnitude change to include the actual pH values.

L. 65: We have added '(from sea to atmosphere)'.

L. 66: We have clarified this to be depth and enclosure differences.

L. 105: We clarified this to be deep pacific water and surface freshwater

L. 108: We have changed this sentence to state 'decrease' rather than 'low'.

L. 110: We clarified that these estimates are specific to aragonite.

L. 118: We do not feel capital P is out of the ordinary (e.g., Hales et al. 2016 and Waldbusser et al. 2015) and have chosen to keep this formatting.

L. 146: We have changed this to 'storm activity'.

L. 186: We have added labels for the two other lagoons.

L. 207: Yes, sediment was collected for these months. The dates are reported in supplementary table 1.

L. 235: We thank the reviewer for providing the additional references here. Lueker over-estimates PCO2 (< 40 uatm) compared to Cai and Wang constants for water masses with low salinity however, these appear to be mostly compensatory approx. -20 uatm with the underestimation of K1 and K2 constants due to low temperatures ( $\sim$ 5–10 C).

We note that the pH NBS scale used for lower salinity waters underestimates pH values relative to Lueker, which incorporates the differences caused by potentiometric measurements and liquid-junction potential at high ionic strength. Given the broad spectrum of salinity values captured during the open phases in this lagoon, there is no one ideal derivation. Quantification of this uncertainty is difficult, however, any potential relative discrepancy would be trivial given the large deviation between duplicate bottle samples.

We have added additional reference to the use of the Lueker constants and acknowledged the potential uncertainties and difficulty in quantifying these. See lines 240–247 in revised manuscript.

L. 279: This refers to properly setting the window, or size of the main lobe (i.e., frequency peak), so the width of the frequency peak does not obscure adjacent frequency peaks.

We have added clarification to this. See line 292 in revised manuscript.

L. 290: Removing points below the freezing line = Removing erroneous data (occurs because something, usually sediments, block the electrical field of the conductivity sensor and conductivity reads erroneously low). We have clarified this in the text.

L. 305: The map has been changed to show these features.

Eq 2: This was following the nomenclature and aesthetics used in Wanninkof 2014 but is not necessary. These have been removed.

End of section 2.5: Please see response to Reviewer # 2 question 2.

Fig 6: We are slightly confused by this comment. The x-axis is salinity and is marked as such on the figure. The color scale is temperature and is also marked appropriately.

In the end: We find that the porewater measurements may be useful for follow-up studies on this system. By presenting them here they become more easily accessible by providing potential benefit to future studies.

Please also note the supplement to this comment:
https://bg.copernicus.org/preprints/bg-2020-358/bg-2020-358-AC1-supplement.pdf
* * *
[Figure]

[Figure]

**Fig. 1.**

---

## Author Comment (AC2) · 22 Dec 2020

We would like to thank both anonymous reviewers for their comments and time to help improve this manuscript.

1. We agree with the review that these are sources of uncertainty that should be given more attention. We have added comments regarding these sources of uncertainty in the revised manuscript. See lines 719–721 in revised manuscript.

While additional sources of uncertainty in AT estimates may be present, we have found similar salinity-AT relationships in arctic waters within the region giving us confidence in supporting the robust correlation we found in our AT measurements.

2. We would like to clarify that this method was chosen because the quantified uncertainty is not identified as an error but are the bounds of the flux potential. That is, quantifying the difference associated between the AT estimates yields a much greater difference than the uncertainty associated with the measurements taken to derive AT.

Since salinity values fluctuate from $\sim$ 6 to 38 during the open phase, setting the freshwater effects on the gas transfer velocity yields a lower bound that is approximately identical to those associated with the measurement uncertainty if taken as a proportion of the total uncertainty calculation for pH and applied to AT estimates.

The difference between the two regressions yields an $\sim$ 200 umol kg-1 difference in AT, so we calculated both estimates and presented the in situ salinity estimates—which yielded the higher value—as the upper bound. As mentioned, if we calculate the measurement uncertainty as a proportion of the derived total pH uncertainty and apply this to the AT estimate, the discrepancy yields a maximum value of 40 umol kg-1. This would result in a trivial flux uncertainty given the bounds applied to the upper limit and nearly identical to the estimated lower bound.

However, we have added additional text to clarify this including the uncertainty associated with the measurements if applied as a proportion to the AT estimate. See lines 332–337 in the revised manuscript.

3. We agree with the reviewer and feel we can discuss the difference between open seasons in a bit more detail. We have addressed these comments in lines 521–528 in the revised manuscript.

In addition, please see our response to Reviewer # 1 comment "b".

4. We have included the PCO2 figure as supplementary figure 3 and moved previous fig. S3 to S4.

Figure 1. We have used a new base map and labeled the adjacent lagoons.

Figure 4,6, 7: The legends and figure captions have been changed to "closed".

Figure 7: We have added gridlines to figure 7.

Figure 8: Detrended pH has been added to the caption

Table 1 and S1: We have added the definition to "*" in table 1, which is the calibration bottle sample. We have added the definition of "*" to table S1 which associates the "*" time and date to the "*" values. This indicator was added since multiple timestamps exist for surface and bottom rows.

Figure S2: Another panel with raw salinity data was added to this figure.

L. 202: This has been changed to August 2018, April 2019, June 2019

L. 203: We have provided examples such as temperature and conductivity.

Section 2.3: We have changed the section title to the one suggested by the reviewer: "Seawater chemistry and pH sensor calibration".

L. 230: We have changed this sentence to clarify we are referring to all benchtop salinity measurements. These salinity measurements refer to the discrete bottle samples collected and reported in table S1.

L. 234: All temperature measurements were recorded from the SeaFET thermistor. We amended this sentence to reflect this.

L. 300: We have changed R2 to R2.

L. 318: We have corrected this to "Wanninkhof (2014)".

L. 329-330: The detrend was performed by subtracting the mean of the best fit line. The purpose of performing a detrend here was to account for potential correlation with other parameters we did not measure in this study. Please see lines 352–355 in the revised manuscript as well as our response to reviewer 1 comment "a".

Section 2.6: We have edited the title of this section and some of its content due to our responses to reviewer 1 comment "a" and reviewer 2 comment: 329-330. We believe

the title and content of this section better describe its utility and placement.

L. 408: We have changed the p-value to < 0.001.

L. 482: We have amended this sentence to indicate "instances and the running average of pHT were found to be > 8.05 across the salinity range from 5 to 30".

L. 561-563: We have amended this sentence and the one that follows to clarify this point. See lines 595–599 in revised manuscript.

L. 572-582: This is difficult to quantify. If we assume our linear regression between salinity and AT is the same between salinity and DIC due to cyroconcentraion, derivations result in an $\sim$ 0.08 pH unit decrease where the ratio between AT:DIC is 1.0217. However, this does not account for the potential of ikiate formation which would modify the AT:DIC at a 2:1 ratio. To observe the decrease in pH found in this study, we would need to observe an AT:DIC ratio of 0.985 which cannot be completely resolved from ikaite precipitation due to saturation state thermodynamics.

If we assume a steady $NH_4^+$ concentration during the open season and steady accumulation from remineralization, the change between August 2018 (noting that closed began $\sim$ 21 October 2018) to April 2019 $NH_4^+$ at the surface is equal to $\sim$ 20 ug L-1. Assuming a stoichiometric relationship between N and C to be 16:106 this would be equal to $\sim$ 130 umol kg-1 increase in DIC over this period. While the $PO_4^{3-}$ values are a bit anomalous, we note that these values depend on the N:P ratio in the remineralized OM and sediment flux of solutes.

We have amended this section to reflect this response including the stoichiometric approximations. See lines 615–625 in the revised manuscript.

L. 600-606: There is evidence for low concentrations of oxygen and low pH to limit nitrification rates (See Middleburg and Levin 2009 and Laverock et al. 2017). We believe our discussion regarding this is feasible, particularly given the high concentrations of ammonium found in the bottom waters and potential of oxidation of the reduced

nitrogen when O2 levels increase.

Please also note the supplement to this comment:
https://bg.copernicus.org/preprints/bg-2020-358/bg-2020-358-AC2-supplement.pdf

---

## Author Response (AR1)

We would like to thank both anonymous reviewers for their comments and time to help improve this manuscript.

**{1} Response to anonymous reviewer 1 marked in bold typeset.**

1 Overview

In this study, Miller et al. describe an entire calendar year of pH, temperature and salinity changes in the Kaktovik Lagoon, Alaska. Using time series of current speed and photosynthetically active radiation, they look for the mechanisms driving these pH changes. They then go further and, using wind speed and atmospheric pCO2 measurements, provide estimates of CO2 fluxes to the atmosphere in the ice-free period. The study reads very well, is scientifically interesting, novel, and should be published given addressing a few points detailed below.

2 Major points

a. The pH dependency on salinity and temperature could be quantified more rigorously than with the present regression. As is, it is unclear how exactly this regression was performed, why it was done like that, etc. The approach of Hagens and Middelburg (2016), who wrote a paper on how to attribute pH variability to governing factors, seems more robust and I would encourage the authors to look in this direction. Looking at Fig. 2, it is hard to believe that salinity and pH are not correlated in the first open phase, as stated on line 372. What was the rationale for applying a 7-day running average on the pH time series? Why not doing it also on the salinity and temperature time series and look for correlations in the smoothed time series instead of in the noisy ones?

**We thank the reviewer for this comment and agree there is a utility in scrutinizing the pH-salinity relationship more vigorously. The approach followed by Hagens and Middleburg 2016 derives pH sensitivities as affected by acid-base constituents along with their dependence of speciation on temperature, salinity, and pressure. The relationship of pH to temperature and salinity in this respect would be due to the modulation of dissociation constants and, thus, carbonate chemistry speciation, if we assume this to be the dominant acid-base dynamic in the lagoon. Hofmann et al. 2009 examined the sensitivity of dissociation constants of acid-base speciation to the concomitant effects on proton concentration. They showed that the main driver of pH was the ratio of $DIC:A_T$ and a function of conservative mixing. Secondary was biological activity. In this paper, we examined the relationship between pH and salinity as a linear quadratic. Which is an appropriate approach given the data we have collected.**

**As mentioned above, quantifying the effects of temperature and salinity on the dissociation constants of the carbonate system and how this modifies pH would be based on using $A_T$ data estimated from our $A_T$-salinity regression, thus resulting in a circular approach at determining pH sensitivity to temperature and salinity. For this reason—and as suggested by the reviewer— we find the approach of examining the relationships between running**

average data more appropriate. Further examination we feel is beyond the scope of this manuscript given the other findings and observations presented.

We have amended the manuscript to reflect our use of this linear relationship approach between pH and salinity and temperature. See lines 349–351, 396–398, 404–405 in the revised manuscript.

We have revised the manuscript to reflect the new findings and make clear that no linear relationships exist with temperature according to our analyses, nor did we find a relationship between salinity and pH during open phase 2018.

Below is the relationship between the 7-d running average for salinity and pH during open phase 2018

[Figure]

We encourage the reviewer to examine figure 3 rather than figure 2 when interpreting the relationships between salinity and pH as figure 3 is a magnified perspective of this data. From this viewpoint, it is more clear that the variability between salinity and pH is non-congruent.

We choose a 7-day running average of the time series because this appeared to give a smoothed line trend that still highlighted small shifts to new baseline pH levels.

New relationships between salinity, temperature, and pH were examined and 7-day running averages were applied to salinity and temperature as well for the times series figures.

b. The authors should go further in the interpretation of the CO2 flux estimates. Clearly, PAR is not correlated with CO2 flux estimates, and something else than biology must govern the high CO2 flux variability. Looking at Fig. 9, it seems that changes in CO2 fluxes are caused by storms and other weather events, rather than biological processes. What about adding current speed (from supp. Fig. 1) on Fig. 9? It would give a better idea whether weather events induce are translated into water turbulence or not. In a shallow lagoon (4.4 meters max) such as that, storms would likely resuspend sediments, releasing DIC in the bottom waters and lead to short term CO2 efflux from the lagoon to the atmosphere. Measuring transmissivity would have helped to quantify sediment resuspension through time. The differences between the CO2 flux and the pH time series, in terms of behavior and possible controlling mechanisms, should be emphasized more in the discussion, abstract and possibly title.

**We find the reviewer's suggestion beneficial and agree that we can discuss the potential mechanisms of $CO_2$ flux variability to a greater depth. We note though that since we, unfortunately, do not have additional data such as transmissivity (which would be beneficial), most of our conclusions are speculative.**

**We had not thought to correlate flux with current speed, however, when reexamined we found no correlation between the two and feel it would not add more to the current figure as is.**

**We found supporting evidence in the literature regarding the reviewer's comment re upwelled $CO_2$. We have incorporated these points into the discussion. See lines 676—683 in revised manuscript. Briefly, we highlight that previous studies (Åberg et al. 2010) have found that storm events create an upwelling of $CO_2$ and erosion of stratified layers and conclude that upwelled $CO_2$ to the surface layers is greater than the $CO_2$ loss to the atmosphere. These findings support our conclusions and we feel have improved our conclusions regarding the disequilibrium being the driving factor of $CO_2$ off gassing.**

**Given the already lengthy abstract we feel that additional commentary on $CO_2$ flux would not entice readers. However, we changed the title to reflect our conclusions regarding $CO_2$ flux.**

c. Doing the integral over time of the CO2 flux to/from the atmosphere would allow to put a nice number on the source or sink behavior of this lagoon. Given that, another point to consider for discussion would be: as the ice extent drops in the next century due to temperature increase, how could that affect CO2 air-water fluxes in these lagoons?

**We have added this additional value of integrated C flux over the entire calendar year which provides a better metric for comparison to other published literature. This value is $1.4 \times 10^{-4}$ Tg C yr$^{-1}$. In addition, we have added the suggestion by the reviewer to include the potential implications this finding may have as ice-free days increases in the coming decades. See line 705 in the revised manuscript.**

d. The whole frequency analysis and its results were quite unclear to me. Specifically, it would be very helpful to indicate or highlight on Fig. 7 the range of frequencies that are representative of given natural events (e.g., tides, seasons, etc.).

**We have added additional detail regarding the frequencies associated with daily irradiance and tides. See lines 298–299 in revised manuscript.**

3 Minor points

Title: Consider something more meaningful/informative than "non-linear extremes"

**We have changed the title to "The Seasonal Phases of an Arctic Lagoon Reveal the Discontinuities of pH Variability and $CO_2$ flux at the Air-sea Interface"**

L. 56: Define LTER

**We have defined LTER**

L.59: Here and elsewhere: because pH is on a log scale, it is a bit meaningless to simply give the magnitude of a pH change (see the technical note from Fassbender et al., 2020, published recently as a preprint in Biogeosciences). For instance, instead of saying "pH varied with a difference of 0.2 units" without giving the actual pH value before, say "pH varied from 7.8 to 8".

**We have amended this sentence and those similar that only reported a magnitude change to include the actual pH values.**

L. 65: "Efflux" is unclear, specify in which direction it's going. Given the large error bar, we can't actually say that it's going in any particular direction.

**We have added '(from sea to atmosphere)'.**

L. 66: Explain what are the "geomorphic differences"

**We have clarified this to be depth and enclosure differences.**

L. 105: Specify "specific water mass mixing patterns"

**We clarified this to be deep pacific water and surface freshwater**

L. 108: Acidification results in "lower" pH and saturation state, but not necessarily "low"

**We have changed this sentence to state 'decrease' rather than 'low'.**

L. 110: Here and in the next sentence: which calcium carbonate mineral are you talking about?

**We clarified that these estimates are specific to aragonite.**

L. 118: Here and elsewhere: writing pCO2 with a capital P is strange, it is usually written "pCO2"

**We do not feel capital P is out of the ordinary (e.g., Hales et al. 2016 and Waldbusser et al. 2015) and have chosen to keep this formatting.**

L. 146: "Meteorological events" is too vague

**We have changed this to 'storm activity'.**

L. 186: Can you show the Arey and Jago lagoons on Fig. 1?

**We have added labels for the two other lagoons.**

L. 207: Were sediments also retrieved in April, June and August?

**Yes, sediment was collected for these months. The dates are reported in supplementary table 1.**

L. 235: About using the Lueker et al. constants: Dinauer and Mucci (2017) showed that carbonic acid dissociation constants (K1* and K2*) of Cai and Wang (1998) seem to be more adapted to low-salinity environments such as this lagoon. Papadimitriou et al. (2018), Millero et al. (2002) and Sulpis et al. (2020) all obtained K1* and K2* values that are lower than those from Lueker et al. (2000) in cold waters such as those from the present lagoon. Each of these estimates, including those from Lueker et al., come with associated uncertainties, but what is rarely taken into account is the uncertainty associated with the choice of constants that one choses for an analysis, because it's very hard to quantify. This additional "hidden" source of uncertainty should be discussed here, especially given how extreme this lagoon is in terms of temperature and salinity.

**We thank the reviewer for providing the additional references here. Lueker overestimates $PCO_2$ (< 40 uatm) compared to Cai and Wang constants for water masses with low salinity however, these appear to be mostly compensatory approx. -20 uatm with the underestimation of K1 and K2 constants due to low temperatures (~ 5–10 C).**

**We note that the pH NBS scale used for lower salinity waters underestimates pH values relative to Lueker, which incorporates the differences caused by potentiometric measurements and liquid-junction potential at high ionic strength. Given the broad spectrum of salinity values captured during the open phases in this lagoon, there is no one ideal derivation. Quantification of this uncertainty is difficult, however, any potential relative discrepancy would be trivial given the large deviation between duplicate bottle samples.**

**We have added additional reference to the use of the Lueker constants and acknowledged the potential uncertainties and difficulty in quantifying these. See lines 239–246 in revised manuscript.**

L. 279: What is a "sidelobe attenuation"?

**This refers to properly setting the window, or size of the main lobe (i.e., frequency peak), so the width of the frequency peak does not obscure adjacent frequency peaks.**

**We have added clarification to this. See line 290 in revised manuscript.**

L. 290: I didn't get "Measurements identified as below the freezing point of water"

**Removing points below the freezing line is equal to removing erroneous data (occurs because something, usually sediments, block the electrical field of the conductivity sensor and conductivity reads erroneously low). We have clarified this in the text.**

L. 305: Can you show the Barrow and Barter Island airport on the map in Fig. 1?

**The map has been changed to show these features.**

Eq. 2: what are the brackets around U2 for?

**This was following the nomenclature and aesthetics used in Wanninkof 2014 but is not necessary. These have been removed.**

End of section 2.5: I didn't understand what the upper/lower bound uncertainties are and what is the link with Eq. 1

**Please see response to Reviewer # 2 question 2.**

Fig. 6: replace y and x by the actual variable names. Are x1 and x2 both salinity?

**We are slightly confused by this comment. The x-axis is salinity and is marked as such on the figure. The color scale is temperature and is also marked appropriately.**

In the end, what were the porewater measurements for?

**We find that the porewater measurements may be useful for follow-up studies on this system. By presenting them here they become more easily accessible by providing potential benefit to future studies.**

4 References

Cai, W.J., Wang, Y., 1998. The chemistry, fluxes, and sources of carbon dioxide in the estuarine waters of the Satilla and Altamaha Rivers, Georgia. Limnology and Oceanography 43, 657-668.

Dinauer, A., Mucci, A., 2017. Spatial variability in surface-water $pCO_2$ and gas exchange in the world's largest semi-enclosed estuarine system: St. Lawrence Estuary (Canada). Biogeosciences 14, 3221-3237.

Hagens, M., Middelburg, J.J., 2016. Attributing seasonal pH variability in surface ocean waters to governing factors: Governing factors of seasonality in pH. Geophysical Research Letters 43, 12528-12537.

Lueker, T.J., Dickson, A.G., Keeling, C.D., 2000. Ocean $pCO_2$ calculated from dissolved inorganic carbon, alkalinity, and equations for K1 and K2: validation based on laboratory measurements of $CO_2$ in gas and seawater at equilibrium. Marine Chemistry 70, 105-119.

Millero, F.J., Pierrot, D., Lee, K., Wanninkhof, R., Feely, R., Sabine, C.L., Key, R.M., Takahashi, T., 2002. Dissociation constants for carbonic acid determined from field measurements. Deep Sea Research I 49, 1705-1723.

Papadimitriou, S., Loucaides, S., Rérolle, V.M.C., Kennedy, P., Achterberg, E.P., Dickson, A.G., Mowlem, M., Kennedy, H., 2018. The stoichiometric dissociation constants of carbonic acid in seawater brines from 298 to 267 K. Geochimica et Cosmochimica Acta 220, 55-70.
Sulpis, O., Lauvset, S.K., Hagens, M., 2020. Current estimates of K1*and K2* appear inconsistent with measured $CO_2$ system parameters in cold oceanic regions. Ocean Sci 16, 847-862.

**NA**

{2} **Response to anonymous reviewer 2**

The authors present the results of annual pH measurements in an Arctic lagoon. They found that salinity-pH relationship differs between observations periods, and that the lagoon acts as $CO_2$ source to the atmosphere during the open season. These observations and findings are interesting and valuable. I recommend that this paper be published in Biogeosciences with minor revisions.

Specific comments:

1.  AT is estimated from S- AT regression line based on 3 samples and is used to calculate $CO_2$ flux. Possible errors in $CO_2$ flux are estimated by comparing estimations using Ssensor-AT and Sbottle-AT. However, S change due to mixing with seawater, mixing with river water, melting and formation of sea ice can change S-AT relationship. This is of particular concern during the melting season (open 2019) when no AT sampling was made. Also, high concentration of organic matter in some coastal regions can significantly contribute to AT. This may be important for the lagoon system. These uncertainties should be discussed in the text.

We agree with the reviewer that these are sources of uncertainty that should be given more attention. We have added comments regarding these sources of uncertainty in the revised manuscript. See lines 719–721 In revised manuscript.

While additional sources of uncertainty in $A_T$ estimates may be present, we have found similar salinity-$A_T$ relationships in arctic waters within the region giving us confidence in supporting the robust correlation we found in our $A_T$ measurements.

2. Also about the error in CO2 flux. I do not understand why uncertainty associate with freshwater Sc and AT-S regression should represent lower and upper bound? How about additive errors?

We would like to clarify that this method was chosen because the quantified uncertainty is not identified as an error but are the bounds of the flux potential. That is, quantifying the difference associated between the $A_T$ estimates yields a much greater difference than the uncertainty associated with the measurements taken to derive $A_T$.

Since salinity values fluctuate from ~ 6 to 38 during the open phase, setting the freshwater effects on the gas transfer velocity yields a lower bound that is approximately identical to those associated with the measurement uncertainty if taken as a proportion of the total uncertainty calculation for pH and applied to $A_T$ estimates.

The difference between the two regressions yields an ~ 200 umol kg$^{-1}$ difference in $A_T$, so we calculated both estimates and presented the *in situ* salinity estimates—which yielded the higher value—as the upper bound. As mentioned, if we calculate the measurement uncertainty as a proportion of the derived total pH uncertainty and apply this to the $A_T$ estimate, the discrepancy yields a maximum value of 40 umol kg$^{-1}$. This would result in a trivial flux uncertainty given the bounds applied to the upper limit and nearly identical to the estimated lower bound.

However, we have added additional text to clarify this including the uncertainty associated with the measurements if applied as a proportion to the $A_T$ estimate. See lines 331–336 in the revised manuscript.

3. Different pH-S relationship between two open periods is interesting but not clearly explained in the text. Open 2018 is in the fall, when vertical mixing can be enhanced by cooling and storms. Accordingly, air-sea CO2 exchange can result in relatively stable pH (and pCO2) despite the change in salinity. On the other hand, open 2019 is in melting season when stratification, warming, dilution, and remineralization at the bottom can largely change pH. Can this explain the difference?

We agree with the reviewer and feel we can discuss the difference between open seasons in a bit more detail. We have addressed these comments in lines 520–527 in the revised manuscript.

In addition, please see our response to Reviewer # 1 comment "b".

4.  I suggest authors to add a figure showing estimated pCO2 time series. It would help to better understand seasonal variability in carbonate chemistry.

**We have included the PCO₂ figure as supplementary figure 3 and moved previous fig. S3 to S4.**

Minor points:
Figure 1: The station location and text in the picture are hard to see. What about changing to white? Please show locations of Arey and Jago lagoons and Beaufort Sea.

**We have used a new base map and labeled the adjacent lagoons.**

Figures 4, 6, 7: Change "iced" and "ice-cover" to "closed" to be consistent with the main text.

**The legends and figure captions have been changed to "closed".**

Figure 7: It is hard to compare position of peaks among figures. Adding vertical lines for some important frequencies should be helpful (1 and 2 d-1?).

**We have added gridlines to figure 7.**

Figure 8: State "detrented pH" in the caption.

**Detrended pH has been added to the caption.**

Table 1 and S1: Explain what * means. Table S1: Add in situ salinity used for AT-Sin situ regression in the table.

**We have added the definition to "*" in table 1, which is the calibration bottle sample. We have added the definition of "*" to table S1 which associates the "*" time and date to the "*" values. This indicator was added since multiple timestamps exist for surface and bottom rows.**

Figure S2: May be better to show original invalid data. Otherwise, this is exactly the same as Figure 2c.

**Another panel with raw salinity data was added to this figure.**

L202 "April, August and June" : Change to "August 2018, April 2019, June 2019, and August 2019"

**This has been changed to August 2018, April 2019, June 2019.**

L203: Specify "physicochemical parameters"

**We have provided examples such as temperature and conductivity.**

Section 2.3 title: "Seawater chemistry and pH sensor calibration"

**We have changed the section title to the one suggested by the reviewer: "Seawater chemistry and pH sensor calibration".**

L230: "All salinity measurements were . . .." Does this mean "Salinity of all AT/pH samples were..."?

**We have changed this sentence to clarify we are referring to all benchtop salinity measurements. These salinity measurements refer to the discrete bottle samples collected and reported in table S1.**

L234: Which temperature was used?

**All temperature measurements were recorded from the SeaFET thermistor. We amended this sentence to reflect this.**

L300: Superscript "2" of "R2". L318: (Wanninkhof, 2014) should be Wanninkhof (2014).

**We have changed R2 to $R^2$.**

L329-330: I could not understand how pH was detrended for open 2018 period when there was no correlation between pH and S. This may be due to my misunderstanding. Please explain how "detrended pH" was obtained in more detail.

**The detrend was performed by subtracting the mean of the best fit line. The purpose of performing a detrend here was to account for potential correlation with other parameters we did not measure in this study. Please see lines 352–355 in the revised manuscript as well as our response to reviewer 1 comment "a".**

Section 2.6 "statistical applications": With this generic title, it is hard to understand what this section describes about. Information in this section would be better moved to the section 3.3, when this application is needed. In that case, first two sentences can be removed as they are already explained in the section 3.1.

**We have edited the title of this section and some of its content due to our responses to reviewer 1 comment "a" and reviewer 2 comment: 330-331. We believe the title and content of this section better describe its utility and placement.**

L408: p-value <0.000 should be <0.001. It cannot be negative.

**We have changed the p-value to < 0.001.**

L482: It is said that "in the open phase of 2018, pH values were observed to be >8.05 despite the striking range of S from 5 to 30." However in figure 2, pH varies from <7.7 to 8.4 in open 2018 period.

**We have amended this sentence to indicate "instances and running average of $pH_T$ were found to be > 8.05 across the salinity range from 5 to 30".**

L561-563: Please modify the text to state that cryoconcentration of DIC (and AT) itself lowers pH. Readers may misunderstand that cryoconcentration of DIC lowers pH only via ikaite formation.

**We have amended this sentence and the one that follows to clarify this point. See lines 595–599 in revised manuscript.**

L572-582: Some quantitative analysis is needed. According to my test calculation using CO2sys, cryoconcentration of DIC, AT, S cannot explain observed large decrease in pH. From very high nutrient concentration in closed season, remineralization seems to have the largest impact on pH decrease. This effect can be roughly quantified from increase in nutrients with RFR. If possible, please add some quantitative analysis about ikaite formation.

**This is difficult to quantify. If we assume our linear regression between salinity and $A_T$ is the same between salinity and DIC due to cyroconcentraion, derivations result in an ~ 0.08 pH unit decrease where the ratio between $A_T$:DIC is 1.0217. However, this does not account for the potential of ikiate formation which would modify the $A_T$:DIC at a 2:1 ratio. To observe the decrease in pH (at our observed range) found in this study, we would need to observe an $A_T$:DIC ratio of 0.985 which cannot be completely resolved from ikaite precipitation due to saturation state thermodynamics.**

**If we assume a steady $NH_4^+$ concentration during the open season and steady accumulation from remineralization and trivial sediment efflux, the change between August 2018 (noting that closed began ~ 21 October 2018) to April 2019 $NH_4^+$ at the surface is equal to ~ 20 ug L$^{-1}$. Assuming a stoichiometric relationship between N and C to be 16:106 this would be equal to ~ 130 umol kg$^{-1}$ increase in DIC over this period. While the $PO_4^{3-}$ values are a bit anomalous, we note that these values depend on the N:P ratio in the remineralized OM and sediment flux of solutes.**

**We have amended this section to reflect this response including the stoichiometric approximations. See lines 615–625 in the revised manuscript.**

L600-606: Is the winter oxygen concentration of 5 mg/L (∼160 umol/L) low enough to limit nitrification?

**There is evidence for low concentrations of oxygen and low pH to limit nitrification rates (See Middleburg and Levin 2009 and Laverock et al. 2017). We believe our discussion regarding this is feasible, particularly given the high concentrations of ammonium found in the bottom waters and potential of oxidation of the reduced nitrogen when $O_2$ levels increase.**

**{3} Response to Associate Editor**

**Citations have been changed to chronological order.**

**Fig.1 has been modified to include exchange pathways and labeling of 'Alaska' and 'Canada'.**

218: total alkalinity (A_T)

**This has been corrected according to recommendation.**

235: indicate to which constant the citations refer to.

**The constants associated with each citation have been added.**

317: typo, Schmidt

**This typo has been corrected**.

Legend Fig. 4: I suggest to use "closed phase" rather than "Iced phase" for consistency with text.

**This has been changed in the figure legend and caption**.

Fig. 8: the gray line is hard to see.

**We have darkened the gray line**.

Why do you compare daily $pH_T$ with instantaneous PAR rather than with daily PAR averages (Fig. 8)?

**We chose not use daily PAR averages for the figure because the zero values during the night hours obscure the variability by collapsing the instantaneous PAR measurements. The calculated correlation coefficients between daily average pH and PAR are the ones presented in the text though. This is clarified in section 2.7.**

488: typo, sensor

**This has been removed in response to a comment by Stephen Gonski on the discussion board.**

578: typo, Amundsen

**This typo has been corrected**.

593: Biogeosciences recommends using molar units: express $O_2$ in umol $L^{-1}$

**This has been changed here and in table S1.**

637-638: "defined as waters north of 70 °N and 100 °W" North of 100 °W?

**This has been corrected to state 'west of 100 °W'.**